# Brains over beauty: A preregistered test of the effects of objectification on women's cognitive performance

**Anne Zola** [1]*, **Renee Engeln** [2]

**1** Department of Medical Social Sciences, Psychometrics and Statistics Core, Northwestern University Feinberg School of Medicine, Chicago, Illinois, United States of America, **2** Department of Psychology, Northwestern University, Evanston, Illinois, United States of America

☯ These authors contributed equally to this work.
* anne.zola@northwestern.edu

**Data Availability Statement:** All data are available in a public repository, https://osf.io/7n34k/?view_only=aeb19dcedac24b0ba8c4998f7a794af6.

## Abstract

The present study was a preregistered, well-powered experimental test of findings related to the effect of state self-objectification and anticipation of the sexualized male gaze on women's cognitive performance. College women ($n = 407$) performed a working memory task in one of three randomly assigned conditions. In the experimental conditions (self-objectification and male gaze), women completed the task while being video recorded from the neck down. In the male gaze condition, participants were told their videos would later be evaluated by men as part of a separate dating study. Women in the control condition were not video recorded. Results indicated women experienced a moderate increase in state self-objectification in both experimental conditions. However, compared to the control condition, women in the experimental conditions did not show reduced performance on the working memory task (in either latency or accuracy), decreases in body satisfaction, or increases in negative mood. Across conditions, state self-objectification was not associated with accuracy or latency on the working memory task. Mixed findings concerning objectification's effect on cognitive performance may be attributed to variability in experimental manipulations and dependent variables employed in this area of research.

## Introduction

In 1997, Fredrickson and Roberts proposed Objectification Theory [1], a detailed treatise on how experiences of sexual objectification (and the self-objectification these experiences can trigger) shape women's psychology. One of the theory's key tenets states that when a woman is objectified, part of her conscious attention is usurped by body monitoring. A year after the theory's publication, this premise was formalized and experimentally tested by Fredrickson et al. [2], who assigned women to try on either a bathing suit or a sweater and then complete a difficult math test while wearing the relevant article of clothing. The authors reported that women in the swimsuit condition performed significantly worse on the math test than women in the sweater condition, concluding that self-objectification interrupts women's cognitive

**Funding:** This research was supported by the Benton J. Underwood Summer Research Fellowship awarded by the Northwestern Psychology Department and an Academic Year Research Grant awarded by Northwestern University's Office of Undergraduate Research (both awarded to AZ).

**Competing interests:** The authors have declared that no competing interests exist.

performance by diverting attentional resources to body monitoring. Results of additional experimental tests of the proposed effect of objectification on women's cognitive performance have been inconsistent (for a review see Winn & Cornelius [3]) and frequently substantially underpowered.

In the current study, we used a well-powered, preregistered experiment to test the impact of both self-objectification and anticipation of a sexualized male gaze on women's performance on a working memory task. Although previous work on this topic has varied in terms of the cognitive processes assessed, we aimed to closely align our study with the theoretical claim that self-objectification interrupts women's conscious attention. We employed a dependent variable designed to test attention and working memory, rather than a specific domain of knowledge (as in previous studies that used advanced math test scores as outcome measures). Moreover, using two experimental conditions, we directly compared the effects of two commonly used objectification manipulations (interpersonal objectification in the form of an anticipated sexualized gaze vs. non-interpersonal objectification that can heighten appearance focus). Additionally, we investigated the effect of these manipulations on women's body satisfaction and mood.

## An overview of objectification theory

According to objectification theory [1], women are frequently the targets of sexual objectification. When objectified, women are treated simply as bodies or body parts to be used for the pleasure of others—an experience that is linked with multiple negative psychological outcomes [4, 5]. The most ubiquitous form of sexual objectification is the visual inspection and evaluation of a woman's body [6, 7]. Sexual objectification in this form can lead to increased body surveillance, body shame, and disordered eating [4, 8–10].

Due to the dominance of heterosexuality in culture and media, the sexualized inspection of women's bodies often takes the form of a *male gaze*. Consistent with arguments by Fredrickson and Roberts [1] that a sexualized gaze can be "real or imagined, present or anticipated" (p. 180), researchers have found that even anticipating the male gaze can increase appearance concerns among women. For example, when told they would be interacting with a male participant, women reported increased body shame and appearance anxiety [11]. The male gaze, like other forms of sexual objectification, can lead a woman to *self*-objectify, or come to view herself as an object to be evaluated by others [1, 12, 13].

Though women's tendency to self-objectify can be conceptualized as a relatively stable, trait-like variable [9], women can also experience temporarily heightened self-objectification (i.e., *state* self-objectification) in response to environmental cues or experimental manipulations. A number of experimental manipulations can increase women's self-objectification, including trying on a bathing suit [2], the presence of mirrors, scales, and fashion magazines [14], and appearance-focused motivational comments by fitness instructors [15]. State self-objectification often manifests as increased body monitoring, known as body surveillance [4].

The initial formulation of objectification theory linked self-objectification to several key psychological outcomes; most of these links have been well studied. For example, substantial research has demonstrated associations between self-objectification and body shame, appearance anxiety, and disordered eating [8, 10, 16, 17]. Research also links state self-objectification with more general negative emotional states. For example, women high in trait self-objectification score higher on measures of anxiety and depression [16]. Moreover, women experience negative emotions like guilt and shame when put in a potentially objectifying context (e.g., trying on a bathing suit, [2, 18, 19]). One study found that simply imagining going out in public in a bathing suit increased women's negative mood [20].

## Self-objectification and cognitive resources

In addition to outcomes related to negative emotionality and disordered eating, Fredrickson and Roberts [1] argued that self-objectification has significant consequences for women's cognitive resources. Specifically, they posited that allocating attentional resources to one's appearance would disrupt other thought processes, explaining that, as a consequence of self-objectification, "significant portions of women's conscious attention can often be usurped by concerns related to real or imagined, present or anticipated, surveyors of their physical appearance" (p. 180). In 1998, Fredrickson and colleagues elaborated upon this premise, suggesting that self-objectification could lead to "diminished performance on any demanding concurrent activity, whether mental or physical." ([2]p. 272). The proposed link between self-objectification and cognitive performance has received less research attention than other areas of objectification theory. Below, we review eleven experimental studies that manipulated either self-objectification or exposure to/anticipation of the male gaze (Table 1). Each study employed either a math test or other cognitive task as a dependent variable (DV). In the spirit of psychology's growing concern about replicability, we pay special attention to issues of statistical power

**Table 1. Key characteristics of eleven experimental studies investigating effects of self-objectification/sexualized male gaze on women's cognitive performance.**

| Author(s) | Manipulation | Outcome measured | Key Result(s) |
|---|---|---|---|
| Fredrickson et al. (1998) | [a]Swimsuit vs. sweater (Took math test while wearing swimsuit/sweater) | Time-limited advanced math test (20 items, 15 minutes) | Women in swimsuit condition performed worse on math test than women in control, $p = .056$ |
| Gapinski et al. (2003) | [ab]Four conditions: swimsuit vs. sweater, "fat talk" vs. no "fat talk" conditions | Time-limited advanced math test (20 items, 10 minutes) | No effect of condition on performance, $p > .05$ (exact test not reported) |
| Hebl et al. (2004) * | [a]Swimsuit vs. sweater (Took math test while wearing swimsuit/sweater) | Time-limited advanced math test (10 items, 10 minutes) | Men and women performed worse on math test in swimsuit condition compared to control, $p < .01$; no significant interaction with gender, $p = .21$ |
| Quinn et al. (2006) | [a]Swimsuit vs sweater (Completed modified Stroop task while wearing swimsuit/sweater) | Modified Stroop task[c] | Women performed worse on task in swimsuit condition compared to control, $p = .039$ |
| Tiggemann & Boundy (2008) | [ab]Objectifying versus non-objectifying environment; then either received an appearance compliment or no compliment | Tests of logical reasoning and spatial orientation | No effect of condition on performance (exact test not reported) |
| Gay & Castano (2010) | [b]Videoed by male vs. female experimenter | Time-limited GMAT test (15 minutes, Study 1); Letter Number Sequencing (LNS) task (Studies 1 and 2) | No effect of condition on math performance, $p = .069$; Significant three-way interaction between condition, TSO, and item difficulty on LNS latency in Studies 1 and 2, $PS < .001$ |
| Gervais et al. (2011) | [t]arget of objectifying gaze from male confederate vs. control (no gaze) | Time-limited advanced math test (12 items, 10 minutes) | Women in objectifying gaze condition performed worse on math test compared to control, $p = .011$ |
| Aubrey & Gerding (2015) | [a] Watched high versus low sexually objectifying music videos | Tests of cognitive resource allocation, encoding, and storage | Women in high sexual objectification condition performed worse on encoding than control, $p < .001$; no effect of condition on resource allocation or storage |
| Guizzo & Casino (2017) | [b]Photographed from the neck down by male vs. female experimenter | Sustained Attention to Response Task | No effect of condition on performance (exact test not reported) |
| Kahalon et al. (2018) * | [b]Received appearance compliment vs. control (no appearance compliment) | Time-limited advanced math test (20 items, 30 minutes) | Men and women performed worse in appearance compliment condition compared to control (controlling for gender, trait self-objectification, and previous test scores), $p = .016$; interaction by gender not reported |
| Savage & Couture Bue (2023) * | [a]Camera on vs. camera on without self-view vs. camera on with self-view | Time-limited advanced math test (9 items, 15 minutes) | No effect of condition on state self-objectification; women with camera on without self-view performed worse than women with camera off |

* Study likely had adequate statistical power for the main effects tested

[a] Indicates manipulation of heightened appearance focus

[b] Indicates manipulation of interpersonal objectification (from which an increase of state self-objectification can ensue)

[c] Dependent variable was calculated as participants' reaction times rather than interference scores

in the review of these studies (most of which were conducted prior to psychology's increased focus on replicability and open science [21]). Though some of these studies included men [2, 22–24], only the results for women are discussed below. For studies in which the authors did not report results separately for men and women and did not report an interaction between gender and condition, we report the results collapsed across participant gender. If the authors did not report exact $p$-values but provided $F$-test values, we calculated $p$-values.

**Studies using a bathing suit as the key manipulation.**   Four studies [2, 19, 23, 25] manipulated women's self-objectification by asking them to try on a bathing suit vs. a sweater. Each of these studies used a modified version of the Twenty Statements Test (TST, as reported in Fredrickson et al. [2]), which was coded for responses that referenced body shape/size or physical appearance. Each study found that trying on the bathing suit made women more likely to provide appearance-related responses on the TST, which the authors interpreted as evidence of self-objectification. Three of these studies [2, 19, 23] used scores on an advanced, time-limited math test as the DV, with problems based on either the Graduate Management Admission Test (GMAT) or the Graduate Record Exam (GRE); the other [26] used performance on a modified Stroop test. Two of the four studies [23, 26] reported significant effects (using the traditional cut-off of $p < .05$) of the swimsuit manipulation on cognitive performance, such that participants scored worse on the math test in the swimsuit condition. Fredrickson et al. [2] reported a $p$ value of .056 for the effect of condition on women's math scores. Gapinski et al. [19] found no effect of the swimsuit manipulation (exact test statistic not reported).

Sample sizes varied widely for these studies (from 42 women to 224 women). Based on current understandings of statistical power (e.g., [27]), three out of the four bathing suit studies reviewed here [2, 19, 26] were likely considerably underpowered for key analyses, which could have contributed to the inconsistency of statistically significant findings. Additionally, the interpretation and reporting of $p$-values differed. Fredrickson et al. [2] and Quinn et al. [26] concluded there was a significant effect of condition with $p$-values of .056 and .039, respectively, whereas Gapinski et al. [19] reported no significant effect of the swimsuit manipulation ($p > .05$, exact test not reported). Studies also varied in terms of the covariates included when testing for a main effect of objectification and it is unclear which covariates were specified a priori vs. examined when the hypothesized main effect was not statistically significant, a common practice among researchers [28].

**Studies using environmental or interpersonal manipulations.**   Tiggemann and Boundy [14] manipulated self-objectification (again assessed by the modified TST) by putting participants in a room with either objectifying cues (mirrors, fashion magazine images, bathroom scales) or a neutral room. Additionally, some participants received an appearance-related compliment from a female experimenter. Using tests of logical reasoning and spatial orientation as DVs, they reported no statistically significant main effects or interactions (using a 2 x 2 x 2 ANOVA with a sample of 96 women; specific $F$-tests and $p$-values not reported). However, this study was severely underpowered (with an average of only 12 participants per cell).

In another study, Aubrey and Gerding [29] induced state self-objectification (measured by the modified TST) by having participants watch either low or high-sexually objectifying music videos. They then tested participants' abilities to process subsequent unrelated advertisements in terms of resource allocation, encoding, and storage. Participants in the high sexual objectification condition performed worse on a visual recognition task of the advertisements than participants in the low sexual objectification condition ($p < .001$, calculated Cohen's $d = 1.23$), which the authors cited as evidence of diminished ability to encode the advertisements. There was no effect of condition on participants' abilities to allocate resources or store information.

In a study that included an a priori power analysis and achieved a power level of .80, Kahalon et al. (Study 2, [24]) manipulated self-objectification via an appearance-related

compliment. When controlling for gender, trait self-objectification, and scores on a math test modeled after a college entry exam, the authors reported a statistically significant negative effect of the compliment on scores on a GRE-based math test ($p = .016$ collapsing across participant gender, total $n = 148$).

In a recent study, Savage and Couture Bue [30] examined the effects of virtual classroom camera settings on state self-objectification (measured by the TST), appearance anxiety, and cognitive performance (assessed by a 15-minute GMAT math test). Although the authors found no significant difference in state self-objectification or appearance anxiety between camera settings, participants who had their camera on without self-view performed worse on the math test than those with their camera off. Results of a moderated mediation analysis suggested state self-objectification could not explain the relationship between camera settings and cognitive performance. The authors included a sensitivity analysis indicating their sample of 167 could detect a small to medium effect ($f = .24$) at a .80 power level.

Other studies more directly manipulated the male gaze, either by having a male (vs. female) experimenter video record/photograph the participant or by having a male confederate gaze at the participant in a sexual manner. Gay and Castano [31] conducted two studies using two measures of cognitive performance (a letter-number sequencing task and a GMAT-based math test). In Study 1 ($n = 25$ women), women were either videoed by a male experimenter (high-objectification condition) or a female experimenter (low-objectification condition). The comparison between math test scores for these two conditions was not statistically significant ($p = .069$). For time taken to complete the sequencing task, the authors reported one statistically significant 2-way interaction; participants high in trait self-objectification took longer in the high-objectification vs. low-objectification condition, $p = .03$. In Study 2 ($n = 50$), the authors dropped the math test and found a non-significant 2-way interaction between condition and trait self-objectification on the time taken to complete the sequencing task ($p = .076$). The authors also reported a significant 3-way interaction: participants high in trait self-objectification performed worse in the high-objectification condition on difficult items ($p < .001$). Once again, both studies were underpowered by today's more stringent standards; it is also unclear which statistical tests conducted were specified a priori.

Guizzo and Cadinu [32] manipulated the male gaze by having a male (vs. female) experimenter photograph the participant from the neck down. The authors found no overall effect of gaze condition on state self-objectification or a sustained attention task in which participants had to withhold responses (i.e., pressing the SPACEBAR) to a target. No exact test statistics were reported for these tests. With a sample of 107 women, the effect size of interest would need to be at least $d = 0.55$ for this study to have a power level of .80. The authors included a moderated mediation analysis using a trait-level variable, internalization of beauty ideals, measured seven days prior to the experiment. For participants in the experimental condition (i.e., those photographed by a man), higher internalization of beauty ideals predicted decreased flow states and reduced performance on the attention task ($p < .001$). In a different study [22], a male confederate gazed at the participant in a sexualized manner while interacting with her. The authors reported that women in the objectifying gaze condition performed significantly worse on a GRE-based math test than women who did not receive the sexualized gaze ($p = .011$, $n = 67$ women). This study was also substantially underpowered for the effect size found.

Taken together, results from experimental tests of self-objectification and the sexualized gaze on women's cognitive performance have been inconsistent. More recent concerns related to statistical power also raise the question of the results' replicability, as smaller samples increase the likelihood of false positives [33]. We found that eight of the eleven studies reviewed were likely substantially underpowered for the effects tested.

## The current study

We designed the current study to address several weaknesses in the literature examining the effects of objectification on women's cognitive performance. First, few researchers have distinguished between interpersonal manipulations of objectification (e.g., being a target of a sexualized male gaze or anticipating such a gaze, receiving an appearance compliment) and non-interpersonal manipulations that can increase awareness of one's appearance (e.g., trying on a bathing suit, being photographed/video recorded). In the current study, we include both types of manipulations in order to distinguish their effects. We designed our experimental conditions to allow us to test the effect of self-objectification resulting from a heightened appearance focus (from being video recorded) and the effect of self-objectification when this heightened appearance focus was paired with anticipated interpersonal sexual objectification in the form of the male gaze.

Second, although the original argument from Objectification Theory [1] was that appearance monitoring would disrupt women's attention, studies on the cognitive effects of self-objectification have varied a great deal in terms of whether the tasks used as DVs actually assessed attention (vs. a specific domain of knowledge or other cognitive processes). Though it certainly requires attention to complete a difficult math test, because these tasks require higher-level math skills in addition to focus, they are less direct measures of the potentially distracting effects of objectification. Time-limited advanced math tests also make it difficult to disentangle latency and accuracy of responses. In the current study, we used a working memory task modeled after a WAIS-IV subtest [34] that is commonly used to assess attention, focus, and working memory and requires no advanced mathematics skills (only basic addition, subtraction, multiplication, and division). Although the problems are simple (e.g., 46 plus 38), mentally solving even simple arithmetic problems requires substantial working memory resources [35].

Additionally, previous researchers who used difficult math exams as the key outcome variable and found significant effects have acknowledged that stereotype threat may be an alternate explanation for their pattern of results [2, 22, 24]. Stereotype threat describes diminished performance in reaction to negative stereotypes about one's group's abilities (e.g., the belief that women are bad at math, [36]).These effects only tend to emerge for highly difficult tasks in which participants are reaching the limits of their skill (for a review, see Pennington et al. [37]); the working memory task we chose would be unlikely to be vulnerable to stereotype threat effects given that it required only simple computations.

Finally, we made several choices in the design of this study to increase confidence in the replicability of its results. In order to prevent the type of analytic flexibility associated with *p*-hacking [21, 38], we preregistered a detailed description of the method, hypotheses, and planned analyses for this study (see preregistration here). Additionally, because previous studies on this topic have frequently lacked sufficient statistical power for the effect sizes obtained, we recruited a large sample and focused on the effects of the key manipulation rather than conducting complex tests with multiple covariates or moderators. See below for power analysis.

In the present study, we randomly assigned women to one of three conditions (control, self-objectification, or male gaze) and asked them to complete an attention-intensive working memory task. In the self-objectification and male gaze conditions, the women completed the task while being video recorded. Participants in the male gaze condition were additionally led to believe the videos would later be evaluated by men in the context of a dating study. We anticipated that women would report the highest levels of state self-objectification in the male gaze condition where the participant's heightened appearance focus (resulting from being video recorded) was compounded with anticipating interpersonal objectification by men in the fictional dating study.

The current experiment was modeled after previous work that used video recording or photographing from the neck down to simulate an objectifying gaze [31, 32]. However, we made several key modifications to previous designs. First, we use an interpersonal objectification manipulation (i.e., anticipated male gaze) found to increase social physique anxiety [11], a construct that overlaps with body surveillance. We selected this manipulation over a paradigm of using female versus male experimenters because that paradigm showed no main effects on body surveillance [32] or math performance [31, 32] in past work. The lack of main effects in these studies may be explained by low power or evidence that female experimenters can also trigger self-objectification in women in some contexts [14]. Second, we did not have participants watch the video recording of their body (as in Gay & Castano [31]), as we reasoned women more often experience objectifying gazes without literally taking the perspective of a third person after the fact. Third, previous studies did not include a control group that was not videoed or photographed, making it difficult to extricate the effect of an objectification manipulation (e.g., being photographed) versus an interpersonal objectification manipulation (e.g., experiencing an objectifying gaze). We included a third, audio-recording only, control condition as a comparison group for our experimental conditions.

Objectification Theory originally proposed that body monitoring can disrupt women's conscious attention. As noted, self-objectification can be conceptualized as both a state and trait variable [9]. Women who have the tendency to self-objectify (i.e., high trait-level objectification) might be more likely to be distracted by body monitoring. Thus, in addition to effects directly tied to our manipulations, we theorized that regardless of condition, engaging in more body monitoring during the cognitive task would be associated with increased distraction, measured as reduced performance on the working memory task.

In addition to the working memory task, women completed measures of state self-objectification (operationalized as body surveillance during the working memory task), guilt, anxiety, and body dissatisfaction (all variables that have been previously linked with objectification, see Moradi & Huang [4] for a review). In previous studies, manipulating self-objectification resulted in worse mood and body image states [2, 18, 19]. Again, we expected that the anticipation of interpersonal objectification in addition to a heightened appearance focus would lead women in the male gaze condition to experience reduced performance on the working memory task and increased anxiety, guilt, and body dissatisfaction compared to the self-objectification condition (video-only).

**Hypotheses.** H1: As a manipulation check, we predicted that body surveillance would be significantly higher in the experimental conditions (self-objectification and male gaze) compared to the control and significantly higher in the male gaze condition compared to the self-objectification condition.

H2: We predicted response latency (i.e., time taken to answer the questions) would be slower in the experimental conditions compared to the control and slower in the male gaze condition compared to the self-objectification condition. Likewise, we expected that response accuracy (total questions answered correctly) would be lower in the experimental conditions compared to the control and lower in the male gaze condition compared to the self-objectification condition.

H3: We hypothesized an association between body surveillance and performance on the working memory task (both latency and accuracy), such that higher body surveillance during the task would be associated with lower performance across conditions.

H4: We expected that both state anxiety and state guilt scores would increase from pre-test to post-test across conditions (i.e., a main effect of time point). We predicted there would be a significant interaction between time point and condition, whereby the increase in state anxiety

and state guilt scores would be greater in the experimental conditions compared to the control.

H5: We predicted that body satisfaction would be significantly lower in the experimental conditions compared to the control and significantly lower in the male gaze condition compared to the self-objectification condition.

## Method

### Power analysis and preregistration

Before recruiting participants for the study, we used G*power [39] to conduct an *a priori* power analysis for the key between-subjects test for performance on the working memory task (a one-way ANOVA with three groups), with a power level of .80, an alpha of .05, and an estimated effect size of $d = 0.35$ ($f = .175$). This analysis suggested a minimum sample size of 350 women; however, we recruited additional women beyond this number in order to account for potential exclusions. Based on data provided in Fredrickson et al.'s [2] study examining the impact of the bathing suit manipulation on math test scores, we calculated an effect size of $d = 0.61$ (the authors did not provide effect sizes). However, we remained conservative in our estimate of a small to medium effect size to account for the tendency of published effect sizes to be significantly inflated [40]. All hypotheses, methods, and planned analyses were submitted in a time-locked preregistration. Full materials and preregistration, including the script for analyses, are here.

### Participants

A total of 407 women took part in the study. Two hundred and forty-seven were assigned to participate as part of an introductory psychology course and received course credit. An additional 160 participants were recruited via flyers and social media ads and were paid $10 each. After exclusions outlined in the preregistration (see below for details), 376 women remained in the sample. Introductory course and paid samples were comparable across all variables with the exception of age; paid participants were significantly older ($M = 20.01$, SD = 1.52) than those recruited in the introductory course ($M = 18.63$, SD = 0.84), $t(374) = 11.37$, $p < .001$, $d = 1.12$. In the entire sample, participants were between the ages of 18–25 ($M = 19.17$, $SD = 1.34$). The plurality of the sample identified as White/Caucasian ($n = 174$, 46%), with 25% identifying as Asian/Asian American ($n = 94$), 10% as Hispanic/Latinx ($n = 39$), 9% as African American/Black ($n = 32$), 8% as multiracial ($n = 31$), and less than one percent as another racial/ethnic identity ($n = 3$). Three participants ($< 1\%$) chose not to report race or ethnicity. The Northwestern University Institutional Review Board approved this study (STU00207314). Participants provided written consent to participate in the study.

### Measures

**Spielberger State-Trait Anxiety Inventory (STAI).** The STAI [41] assesses state and trait anxiety. Although the most common form of the measure includes 20-items for both state and trait anxiety, we used a short-form 6-item state scale in this study [42]. Respondents are asked to report the extent to which they feel a given emotion "RIGHT NOW, IN THE MOMENT" from 1(Not at all) to 4 (Very much). Sample statements include "I feel calm" and "I am tense." Mean scores are calculated after reverse scoring the appropriate items, with higher scores indicating greater state anxiety. Scores on this six-item scale are comparably sensitive to pre/post changes in anxiety as scores on the full form (e.g., changes in women's anxiety before and after receiving medical test results [42]) Scores on the short-form STAI have also been successfully

used to assess changes in undergraduates' state anxiety before and after experimental manipulations [43], including the bathing suit and sweater manipulation [19]. Marteau and Bekker [42] reported a Cronbach's alpha of .82 in a mixed sample of students and pregnant women. In this sample, Cronbach alphas for pre-test and post-test STAI scores were .81 and .84, respectively.

**Positive and Negative Affect Schedule (PANAS-X), guilt subscale.** The PANAS [44] is a general measure of self-reported positive and negative emotionality. In the current study, we used the six-item PANAS-X guilt subscale [45]. Participants are asked the extent to which they feel a given emotion at that moment, ranging from 1 (Very slightly or not at all) to 5 (Very much) on items such as "Guilty," "Angry at self," and "Dissatisfied with self." Higher mean scores indicate greater state guilt. Pre- and post-test scores on this measure have been successfully used to capture state changes in guilt in experimental settings [46] and using ecological momentary assessment methods [47]. Watson and Clark [45] reported a median Cronbach's alpha of .88 for the guilt subscale scores across 11 samples that included undergraduate students, adults, and clinical populations. In the present sample, Cronbach's alphas were .84 and .86 for the pre-test and post-test, respectively.

**Body Image State Scale (BISS).** The BISS [48] assesses evaluative and affective body image states in response to experimental manipulations. Participants are asked to indicate how they feel about their body image/physical attractiveness at that moment using a series of six fully anchored nine-point response scales (for example, ranging from "**Extremely dissatisfied** with my physical appearance" to "**Extremely satisfied** with my physical appearance"; emphasis original). Mean BISS scores are calculated after reverse scoring appropriate items. We added an additional item as an attention-check, instructing participants to select the "Much worse" option. Higher BISS scores indicate more favorable body image states. Among women, BISS scores show a small to moderate negative correlation with body surveillance and strong associations with other evaluative body-image measures [48]. BISS scores are also sensitive to experimental manipulations (e.g., overhearing "fat talk" [49]; exposure to idealized media images of women [50]). Cash et al. [48] reported alphas ranging from .77 to .90 among women who took the BISS in neutral, negative, or positive imagined contexts. Alpha for the current sample was .80.

**Objectified Body Consciousness Scale (OBCS): Body surveillance.** The OBCS [51] assesses the experience of objectified body consciousness, or experiencing the body as an object and holding beliefs that support this perspective. The OBCS includes three subscales: body surveillance, body shame, and appearance control. Consistent with previous work in this area [15, 52–54], we used scores on the body surveillance subscale as an indicator of state self-objectification, such that higher mean body surveillance scores indicated greater experiences of state self-objectification. The original authors of the scale describe scores on this subscale as assessing "the amount of time a woman spends watching her body as an outside observer" ([51],p. 209). In samples of college women, body surveillance scores have been correlated with increased body shame, public self-consciousness, and state body satisfaction [48] as well as negative body talk [55]. In a large sample of adult women, scores on this measure were correlated with experiences of interpersonal sexual objectification [56].

In its original form, the scale contains eight items about the extent to which a woman agrees with statements about monitoring her outward appearance; the response scale ranges from 1 (Strongly Disagree) to 7 (Strongly Agree). For the current study, questions were adapted as in Engeln et al. [15], except with reference to the "math task" instead of "fitness class." For example, "During the fitness class, I rarely thought about how I looked," was changed to "During the math task, I rarely thought about how I looked." Additionally, we omitted items three and eight from the original measure as they were not appropriate for the context of this study (leaving a total of six items). McKinley and Hyde [51] reported alphas for the original body

surveillance subscale between .76 and .89 for samples of undergraduate and middle-aged women. Engeln et al. [15] reported an alpha of .84 for the adapted scale in a sample of college women. In the current study, alpha for the adapted measure was .81. See here for full list of adapted items.

**Working memory task.** We modeled our working memory task after the Arithmetic subsection of the Wechsler Adult Intelligence Scale-IV (WAIS-IV; [34]), which assesses focus, attention, and working memory. In a sample of college students, scores on this task were highly correlated with scores on an overall Working Memory Index and moderately correlated with scores on other working memory subtests (Letter Number Sequencing and Digit Span; [57]). Detailed psychometric analyses of the WAIS standardization sample indicate that scores on this subscale load on an index of short-term memory [58], which is highly linked to attentional processes and susceptible to distraction [59]. We created similar items to those on this WAIS subscale rather than using a traditional math test with more advanced problems as in Fredrickson and Robert's swimsuit study [2]. This decision was made in order to focus on the attentional resources specified by objectification theory rather than skills more sensitive to previous training and math courses taken. All the problems we created for our working memory task could be solved using only simple addition, subtraction, multiplication, and division. However, as in the Arithmetic subscale of the WAIS, we required participants to solve the problems in their head, without the use of scratch paper. See here (in the document titled MATH appendices) for list of items.

## Procedure

We invited college women aged 18–25 to take part in a study related to "emotion, body language, and simple mental math." The study took place at a large, selective, Midwestern university in the U.S. Participants were randomly assigned to one of three conditions: control, self-objectification, or male gaze. First, each participant completed a short pre-test on an iPad with measures of state anxiety and guilt. The order of the measures was counterbalanced. Next, the experimenter (always a woman) asked the participant to stand on a specific spot marked with an "X." Consistent with the cover story suggesting the study was focused on body language, the experimenter explained that the participant would need to stand for the entire math task. In order to ensure that participants paid close attention to the instructions, we asked participants to read the written instructions aloud. Across conditions, the instructions described how the participant would hear a series of math questions given to them by an audio recording and should solve the problems in their head, writing their final answers on a whiteboard provided to them. Additional instructions varied by condition: women in the control condition were instructed to turn the whiteboard and show their final answers to the experimenter, whereas women in the experimental conditions were instructed to show their answers to a video camera. See preregistration materials for full script.

The instructions for the experimental conditions described how the participant would be recorded with her face out of frame to keep anonymity, and thus, only her body from the neck down would be recorded. Recording only from the neck down, as in Gay and Castano [31], mimics the objectified view of women's bodies often portrayed in media (i.e., headless or faceless). To manage participants' expectations of who would view the video, the final instructions read, "The video may be reviewed by a researcher after the study, but I understand that she will not be able to see my face." The use of feminine pronouns in instructions to signal gender was modeled after the key manipulation in Calogero [11]. In the male gaze condition alone, experimenters then told participants the videos would be used in an additional study "looking at how men evaluate women and what they find attractive in potential dates." Experimenters

assured the participant that the men in that study would not be able to see the participant's face, hear her voice, or know how she performed on the math task. In other words, they would only be viewing a video of her body.

In both experimental conditions (self-objectification and male gaze), the viewfinder of the video camera was temporarily turned in the participant's direction so she could see that her body from the neck down was captured in the camera's lens. The viewfinder remained in the direction of the participant during the practice problems but was turned back toward the experimenter for the duration of the actual test; no participant could see her image during the working memory task. In the control condition, no video camera was present.

In all conditions, participants listened to an audio recording of a woman's voice dictating an arithmetic problem, solved the problem in their head, and wrote their final answer on a whiteboard. After answering each question, the participant turned the whiteboard in the direction of the experimenter or video camera, depending on condition. The participant pressed a button when she was ready to hear the next question and repeated this procedure for all twenty questions. Participants were encouraged to do the problems as quickly and accurately as possible. The experimenter recorded the number of correct responses. To determine how long each participant took to answer all twenty questions, an audio recording was taken in the control condition using an iPad. These audio recordings and the audio from the videos taken in experimental conditions were reviewed after the study. Two independent raters, blind to condition, used only the audio of these recordings to determine the total time from hearing the first question to answering the final question.

Once the working memory task was completed, participants were invited once again to sit down and were given the post-test survey on an iPad. This survey included the measures of state anxiety and guilt, as well as measures of state body surveillance and body satisfaction. The order of the measures was counterbalanced. Participants then provided basic demographic information and were asked to guess the hypothesis and purpose of the study. Finally, experimenters fully debriefed participants of all deception and explained the true purpose of the study.

## Results

### Data cleaning

Ten participants failed the attention check included in the BISS; their data were excluded. We excluded data from 13 participants from the male gaze condition because their response to the hypothesis guessing question indicated either that they did not believe the follow-up dating study was real or that they correctly guessed the true purpose of the study. Data from seven participants were excluded due to experimenter error or failure of the participant to follow directions (e.g., the participant used the whiteboard for calculations instead of just for recording answers; the participant missed a page of the pre-test or post-test). One participant asked to leave the study after reading the instructions. This left 376 participants for analyses; 121 in the control, 136 in the self-objectification, and 119 in the male gaze condition. Of this sample, 51 participants had some degree of missing data for the working memory task. Twenty-four were missing timing data for the working memory task due to a corrupted SD memory card. We retained all other data, including response accuracy, for these participants. Additionally, response accuracy and latency data were excluded for 27 participants due to a technical error on the laptop in which one or more of the audio files that dictated the arithmetic problems were inadvertently skipped; all other data from these participants were preserved.

For all other variables (i.e., pre- and post-test scores), analysis of patterns of missing data revealed that less than .16% of all items for all cases were missing; ninety-four percent of

**Table 2. Descriptive statistics for post-test measures, all conditions.**

|  | Control (n = 121) | | Self-objectification (n = 136) | | Male gaze (n = 119) | |
|---|---|---|---|---|---|---|
|  | M (SD) | 95% CI | M (SD) | 95% CI | M (SD) | 95% CI |
| State self-objectification | 2.25 (1.07) | [2.06, 2.45] | 2.72 (1.32) | [2.49, 2.94] | 2.80 (1.32) | [2.56, 3.04] |
| Body satisfaction | 5.01 (1.51) | [4.73, 5.28] | 4.96 (1.28) | [4.74, 5.18] | 4.91 (1.43) | [4.65, 5.18] |
| Response accuracy | 16.29 (2.71) | [15.79, 16.79] | 16.35 (2.77) | [15.86, 16.83] | 16.44 (2.54) | [15.96, 16.91] |
| Response latency (s) | 327.23 (108.03) | [306.91, 347.55] | 332.04 (83.61) | [316.53, 347.56] | 322.26 (88.24) | [304.75, 339.76] |

participants had no missing data. Finally, no item had 1% or more missing values. Given the low levels of missing data and consistent with recommendations by Parent [60], available item analysis (i.e., pairwise deletion) was used for analyses below. We ran all analyses in R [61] and calculated Bayes factors with JASP [62]. We interpret Bayes factors using guidelines provided in Doorn et al. [63]. Code and output for all analyses are available here. See Tables 2 and 3 for descriptive statistics for all variables.

## Manipulation check

A one-way ANOVA with body surveillance as the dependent variable and condition as the between-subjects variable indicated a moderate, significant effect of condition on body surveillance, $F(2, 371) = 6.77$, $p = .001$, $\eta_p^2 = .035$. The Bayes factor indicated strong evidence for the alternative over the null model ($BF_{10} = 14.47$). Simple effects analyses demonstrated that compared to the control ($M = 2.25$, $SD = 1.07$), body surveillance was significantly higher in the self-objectification condition ($M = 2.72$, $SD = 1.32$), $t(371) = 2.96$, $p = .003$, $d = 0.38$, 95% CI [0.15, 0.77], $BF_{10} = 6.16$. Similarly, body surveillance was significantly higher in the male gaze condition ($M = 2.80$, $SD = 1.32$), compared to the control, $t(238) = 3.40$, $p < .001$, $d = 0.45$, 95% CI [0.22, 0.86], $BF_{10} = 25.77$. However, body surveillance scores in the two experimental conditions did not differ significantly from each other, $t(251) = 0.53$, $p = .59$, $d = 0.06$, 95% CI [-0.23, 0.39], $BF_{10} = 0.16$. In sum, both experimental conditions similarly increased body surveillance, with moderate effect sizes.

## Response accuracy and latency

Number of problems answered correctly ranged from 7 to 20. To analyze the effect of condition on response accuracy, we ran a one-way ANOVA with number of problems correctly answered as the DV and condition as the between-subjects variable. Results suggested no effect of condition on response accuracy, $F(2, 350) = 0.09$, $p = .92$, $\eta_p^2 < .001$. The Bayes factor indicated strong evidence for the null model over the alternative model ($BF_{10} = 0.04$). Time taken to complete the problems ranged from 155 to 741 seconds. A second one-way ANOVA was run with total time taken to complete the problems as the DV and condition as the between-subjects variable. There was not a significant effect of condition on time taken to complete the

**Table 3. Descriptive statistics for pre- and post-test measures, all conditions.**

|  | Control (n = 121) | | Self-objectification (n = 136) | | Male gaze (n = 119) | |
|---|---|---|---|---|---|---|
|  | M (SD) | 95% CI | M (SD) | 95% CI | M (SD) | 95% CI |
| Pre-test anxiety | 2.07 (0.67) | [1.96, 2.20] | 2.00 (0.54) | [1.90, 2.09] | 1.92 (0.49) | [1.83, 2.01] |
| Post-test anxiety | 2.37 (0.66) | [2.25, 2.49] | 2.33 (0.67) | [2.22, 2.45] | 2.24 (0.60) | [2.13, 2.35] |
| Pre-test guilt | 1.41 (0.50) | [1.32, 1.50] | 1.44 (0.58) | [1.34, 1.53] | 1.30 (0.45) | [1.22, 1.38] |
| Post-test guilt | 1.59 (0.67) | [1.47, 1.71] | 1.64 (0.72) | [1.52, 1.77] | 1.55 (0.58) | [1.44, 1.65] |

problems, $F(2, 322) = 0.29$, $p = .75$, $\eta_p^2 = .002$. Again, the Bayes factor suggested strong evidence for the null model over the model with condition (BF = 0.03). In sum, results suggested no effect of condition on women's response latency or accuracy on the working memory task.

We hypothesized that there would be an association between body surveillance and performance, whereby greater body surveillance during the math task (regardless of condition) would be correlated with reduced response accuracy and latency. Pearson correlations revealed no significant association between state body surveillance scores and accuracy, $r(349) = —.08$, $p = .14$, or latency, $r(322) = .06$, $p = .28$. Bayes factors of 0.20 and 0.12, respectively, indicated moderate evidence in favor of the null over the alternative hypothesis. There was a moderate, negative association between time taken to complete the task and number of correct responses, $r(323) = -.32$, $p < .001$, $BF_{10} = 2.4\text{x}10^6$; participants who took longer on the task answered more questions incorrectly.

As exploratory analyses, we examined the associations between body surveillance, response accuracy, and response latency separately for each condition. There were small-to-moderate negative correlations between response accuracy and latency in the control, $r(109) = -.30$, $p = .002$, $BF_{10} = 15.43$, self-objectification, $r(112) = -.27$, $p = .004$, $BF_{10} = 6.76$, and male gaze conditions, $r(98) = -.43$, $p < .001$, $BF_{10} = 2445.66$. There were no significant correlations in any condition between body surveillance and response accuracy (control: $r(119) = -.11$, $p = .17$, $BF_{10} = 0.23$; self-objectification: $r(132) = -.11$, $p = .27$, $BF_{10} = 0.15$; male gaze: $r(117) = -.03$, $p = .77$, $BF_{10} = 0.17$) or latency (control: $r(109) = .16$, $p = .09$, $BF_{10} = 0.48$; self-objectification: $r(111) = .08$, $p = .41$, $BF_{10} = 0.16$; male gaze: $r(98)$, $-.07$, $p = .51$, $BF_{10} = 0.16$). Bayes factors suggested weak-to-moderate evidence for associations between body surveillance and accuracy or latency when calculated for each condition separately. However, these correlations were small and in one case, in the opposite direction of predictions (i.e., greater body surveillance was associated with shorter response times in the male gaze condition).

## Body satisfaction, anxiety, and guilt

A one-way analysis of variance (ANOVA) with state body satisfaction scores as the DV and condition as the between-subjects variable indicated no effect of condition on state body satisfaction, $F(2, 364) = 0.13$, $p = .88$, $\eta_p^2 < .001$. The Bayes factor suggested strong evidence in favor of the null model or the alternative model, BF = 0.03. We ran two mixed ANOVAs to determine the effect of condition on anxiety and guilt scores. Condition was entered as the between-subjects factor and time-point (pre, post) was entered as the within-subjects factor. Results indicated a large, significant effect of time-point on anxiety, Wilk's $\lambda = .78$, $F(1, 369) = 104.81$, $p < .001$, $\eta_p^2 = .22$, and guilt, Wilk's $\lambda = .90$, $F(1, 368) = 42.10$, $p < .001$, $\eta_p^2 = .10$. Both anxiety and guilt scores increased from pre-test to post-test. However, there was no interaction between time-point and condition for anxiety, Wilk's $\lambda = .99$, $F(2, 369) = 0.14$, $p = .87$, $\eta_p^2 = .001$, or guilt scores, Wilk's $\lambda = .99$, $F(2, 368) = 0.42$, $p = .66$, $\eta_p^2 = .002$. Bayes factors indicated moderate evidence in favor of the time-point only model (pre, post) than a model with time-point and condition (anxiety $BF_{10} = 0.31$, guilt $BF_{10} = 0.21$).

As an exploratory analysis, we examined the correlations between body surveillance during the working memory task and body satisfaction. Results indicated a moderate, negative correlation between state body satisfaction and body surveillance, $r(363) = -.38$, $p < .001$, $BF_{10} = 5.9\text{x}10^{10}$. In other words, across conditions, women who reported engaging in more body monitoring during the task reported less satisfaction with their body's appearance. Additionally, we correlated scores for body surveillance during the task with change scores for anxiety and guilt (a positive change score indicated an increase in anxiety or guilt from pre-test to post-test). We found small-to-moderate negative associations between body surveillance

during the task and change scores for anxiety, $r(368) = .25$, $p < .001$, $BF_{10} = 8398.27$, and guilt, $r(367) = .31$, $p < .001$, $BF_{10} = 1.03 \times 10^7$. In sum, higher levels of body surveillance during the working memory task were associated with an increase in negative emotional states and with lower body satisfaction, regardless of condition.

## Discussion

Overall, our results suggest that although videoing women from the neck-down effectively increased state self-objectification relative to an audio-only control (a moderate effect), there was no evidence that this heightened appearance monitoring reduced women's ability to perform a cognitively-taxing working memory task. Moreover, we did not find evidence consistent with previous work demonstrating decreased body satisfaction or increased negative mood (guilt or anxiety) as a result of objectification [2, 18, 19]. However, consistent with previous work linking appearance monitoring to body dissatisfaction [48], body surveillance during the working memory task showed a moderate, negative correlation with body satisfaction scores. In terms of affect, the primary finding from the current study was an increase in anxiety and guilt from pre-test to post-test (i.e., before and after the working memory task) across all conditions. In other words, completing the difficult working memory task simply made participants feel worse, regardless of objectification manipulations. Below, we discuss interpretations of these results in light of the range of independent and dependent variables used in this area of objectification research.

### Experimental manipulations of objectification and self-objectification

Researchers have used a variety of lab-based methods to manipulate women's objectification or self-objectification, each with strengths and weaknesses. We selected our manipulations using several criteria. First, we aimed to distinguish between interpersonal objectification manipulations and manipulations that increase women's appearance focus but do not have an interpersonal element. In previous studies, interpersonal manipulations have included anticipating or being the target of a sexualized gaze or receiving an appearance compliment; non-interpersonal manipulations have included trying on a bathing suit or being photographed/video recorded. We used a non-interpersonal self-objectification manipulation that we believed would increase women's awareness of their own appearance (being videotaped from the neck down), and for our "male gaze" condition added an interpersonal element by telling women their video would later be evaluated by men. We also prioritized a believable experimental scenario; few women were removed from analyses for expressing doubts about the "male gaze" condition's cover story.

As hypothesized, the women in our study experienced a moderate increase in state self-objectification when recorded from the neck down by a video camera. We hypothesized that women would experience even greater state self-objectification when anticipating a man would later evaluate this video for a dating study. The original formulation of Objectification Theory [1] suggested that when women are chronically the target of interpersonal sexual objectification, they internalize the perspective of an outsider observer; in other words, when her appearance is surveyed by others, a woman learns to become a surveyor of her own appearance (i.e., self-objectification). A potent trigger of self-objectification is being objectified by another (in particular, a man). For this reason, we hypothesized that adding the potential for appearance evaluation or sexualization by a man (in the "male gaze" condition) would heighten the state self-objectification triggered by the video camera manipulation alone, which would simply draw a woman's attention to her appearance. However, anticipating this male

gaze did not appear to increase women's appearance monitoring beyond increases already resulting from the video manipulation.

Without a significant difference in state self-objectification between the self-objectification and male gaze conditions, we were unable to explore the potential impact of interpersonal objectification (i.e., an anticipated male gaze) when compounded with an environmental trigger of self-objectification (i.e., being video recorded). Future researchers should distinguish objectification manipulations that result in increased appearance-focus (e.g., the use of cameras or mirrors) from manipulations of interpersonal objectification (e.g., appearance compliments from a confederate) and be clear about which they are attempting to manipulate in their study. Our results may suggest adding an interpersonal form of objectification, like an anticipated male gaze, does not heighten a woman's appearance monitoring beyond environmental scenarios (i.e., being video recorded). It is also possible that our interpersonal objectification manipulation was too weak. More specifically, women may not have experienced the manipulation as sexual objectification because they knew their video would be anonymous (showing them only from the neck down) and knew they would not interact with the men who viewed their video. Future well-powered studies could investigate whether there is a differential impact on women's cognitive performance after being the target of an actual male gaze via a confederate as in Gervais et al. [22] vs. anticipating a male gaze as in Calogero [11].

Further, unlike experiments where an appearance compliment was the key manipulation [14, 24], women in the current study did not receive feedback on their appearance. In consideration of recent experimental findings that receiving an appearance compliment negatively impacted both men's and women's performance on a cognitive task [24], the lack of significant results related to cognitive effects in the current study could suggest that appearance feedback is a key component of interpersonal objectification that leads to cognitive impairment. However, other experiments have shown that women experience increases in body shame and appearance anxiety when anticipating the male gaze, even without direct evaluation of how they look [11]. Many questions remain as to which forms of interpersonal objectification are most likely to impact cognitive performance; future researchers should endeavor to directly compare the effects of anticipated vs. actual male gaze and the impacts of an appearance compliment vs. appearance evaluation sans feedback.

The video manipulation used in this study was selected to mimic how women experience objectification simply by being looked at, absent additional triggers, like a revealing outfit (e.g., a bathing suit [2]) or an appearance-related comment [14, 24]. Being watched by a video camera is arguably more similar to everyday experiences of objectification (e.g., when your body is being looked at or appraised by others) than completing an advanced math test in a room alone while wearing a bathing suit. With the goal of simulating more realistic experiences of objectification, we did not have participants review their video footage as in Gay and Castano [31] or Guizzo and Cadinu [32] (where participants reviewed photographs of their bodies). Further, our participants completed the working memory task while experiencing the objectification manipulation rather than in tests *after* being videoed/photographed. Even though our video manipulation did increase state self-objectification, this manipulation could be critiqued as being too subtle. Although participants in the self-objectification and male gaze conditions experienced significantly greater body surveillance relative to control, average levels of body surveillance in the experimental conditions fell below the mid-point of the scale. However, 20 percent of women in both the self-objectification ($n = 27$) and male gaze conditions ($n = 24$) scored in the upper half of the body surveillance scale (i.e., scores greater than 4.00), compared to only 7 percent ($n = 8$) of women in the control condition. This finding lends credence to the claim that the video manipulation effectively increased body surveillance relative to the control

(audio-only) and is consistent with previous research suggesting that women vary in terms of the extent to which they self-objectify in response to objectifying experiences [14, 16, 64].

Objectification theory proposes that when objectified, a portion of a woman's conscious attention is diverted to monitor her appearance. Thus, we would expect that, regardless of the strength of the manipulation, experiencing greater body surveillance during the working memory task would be associated with greater latency and reduced accuracy on the working memory task. However, we found no association between higher body surveillance and response latency or accuracy within or across conditions.

## Cognitive tasks used as dependent variables in objectification research

We selected a cognitive task for our key dependent variable that differs from those used in other studies. The math tests used in several previous studies assess advanced math skills [2, 23, 24]. We chose a different approach, administering a task that was designed to be more attention-intensive and tax working memory (which is highly linked to the ability to control attention [65]). Fredrickson and Roberts [1] proposed that when women's attention is drawn to their appearance (which the manipulation in this current study successfully did), this can "profoundly disrupt a woman's flow of consciousness" (p. 180). Thus, we anticipated that a task requiring attentional resources more than math expertise would be more sensitive to increases in self-objectification. Modeled after the WAIS-IV [34] Arithmetic subsection, participants completed computationally simple math problems requiring only basic addition, subtraction, multiplication, and division skills. The simplicity of the problems was also intended to avoid stereotype threat effects, which are more likely to emerge on difficult tasks (for a review see Pennington et al. [37]). This was especially important as stereotype threat effects have been raised as an alternate explanation for women's reduced performance on advanced math tests in objectification studies [2, 22, 24]. Despite being computationally simple, our task required participants to listen to, remember, and answer all questions without making additional notes or having the question repeated, a challenge that required attention and focus. Even simple arithmetic operations require substantial working memory and could be disrupted by other cognitive demands [66], such as body monitoring. Moreover, math problems are frequently employed as distractor tasks in experiments because they are cognitively tasking and load working memory (e.g., [67]).

We did not impose a time limit for our cognitive task. Participants were simply instructed to solve the problems "as quickly and as accurately as possible." This allowed us to compare the total time taken to complete the task between conditions, arguably a better measure of distraction than accuracy alone [66]. Consistent with this goal, our results showed substantial variability in the time taken to complete the task, even among participants with perfect accuracy. Nonetheless, we did not specifically pre-test whether scores on this task would be sensitive to experimental manipulations, something future researchers should consider before undertaking additional work using this or similar tasks. Perhaps mirroring the way stereotype threat tends to emerge at the limits of one's ability, objectification may have a greater impact on tasks that primarily test the participant's ability to determine the correct response, rather than a task that primarily loads working memory. However, the conceptualization of objectification theory focused on self-objectification as distracting, suggesting that self-objectification should impair performance on attention-intensive tasks.

It is possible our working memory task created more anxiety than other attention-based tasks used in this area of research (e.g., the modified Stroop task [26]; Letter-Number Sequencing task [31]). Women's anxiety and guilt increased from pre-test to post-test across all three conditions, consistent with findings on women's and girls' math anxiety [68, 69]. Although the

working memory task included only simple arithmetic problems (e.g., 22 plus 39), the inability to control for previous math scores [2, 24] or math identification [70] is a limitation of the current study. The fact that we asked women to produce their answers on the spot, in front of an experimenter, may also have led this task to be more anxiety-provoking than the paper and pencil tasks used in other studies [2, 23]. Nonetheless, exploratory analyses showed only small correlations between anxiety change scores (from pre-test to post-test) and accuracy or latency (*r*s *of* -.19 and .14, respectively), suggesting that anxiety was not the primary influence on women's performance on this task.

The narrowness of the sample in the current study was a key limitation. Participants attended a highly selective university where the average math ACT/SAT score of admitted students is in the 90-98th national percentile. Although the task was designed to task working memory regardless of math expertise and did not show evidence of ceiling effects, the participants likely had higher motivation and capacity to succeed at the task (even under objectifying conditions) than a more varied sample might have. However, scores on the working memory task still showed meaningful variability in terms of both accuracy and latency. Additionally, previous research testing the impact of objectification on cognitive tasks also used samples of college women [23].

## Individual differences

The current study did not examine the role individual differences might play in objectification's effect on cognitive performance. In particular, sexual orientation could influence the extent to which a woman experiences an anticipated male gaze as sexual objectification. However, in their original formulation of objectification theory, Fredrickson and Roberts [1] argued that the sexual objectification of women is so chronic and pervasive that it is likely to affect women regardless of sexual orientation. Indeed, some studies have shown that lesbian women experience similar levels of interpersonal sexual objectification as heterosexual women [56, 71], including the sexualized male gaze [72]. The extent to which women enjoy being sexualized [73] might also play a role in how women react to objectification manipulations. Although body monitoring is theorized to be mentally taxing despite one's enjoyment, women who do not enjoy sexualization could experience larger effects in terms of negative mood.

Though researchers have documented many ways in which women have similar experiences with objectification regardless of racial/ethnic background (see Moradi & Huang [4] for a review), others have pointed to evidence that women of different racial/ethnic backgrounds may experience the effects of objectification differently or at different rates. For example, Black women may be especially likely to be both sexualized and explicitly dehumanized by observers [74, 75], and experiences of objectification may overlap substantially with racial microaggressions [76]. Though the sample for the current study was relatively diverse in terms of race/ethnicity, we did not have sufficient statistical power to test for a potentially moderating role of this variable. Future research on the cognitive effects of objectification should include experiments specifically designed for such tests.

In addition to race/ethnicity and sexual orientation, individual differences in trait self-objectification, or the degree to which one chronically self-objectifies, are important to consider. For example, Kahalon et al. (Study 1, [24]) found that after receiving an appearance compliment, women with high-trait self-objectification performed worse on a math test than those with low-trait self-objectification. In the current study, the inability to evaluate the effects of pre-existing individual levels of self-objectification and body dissatisfaction by assessing these outcomes before the task was a weakness. However, we chose to present body-related measures only after the working memory task to avoid priming participants to think about

their bodies or triggering demand characteristics [77]. Future researchers could avoid these demand characteristics by splitting the study into two parts as in Kahalon et al. [24], wherein participants complete measures of potential moderators and dependent variables in separate sessions.

Overall, the role of trait self-objectification as an important moderator of the effects of objectification on cognitive performance is unclear. Among studies reviewed that did include a measure of trait self-objectification or similar trait-level variable, results are mixed. Five studies [2, 14, 19, 23, 30] found no evidence that trait self-objectification moderates the effect of objectification on performance. Of these, one reported a significant main effect of objectification on performance using the cut-off $p < .05$ [23], four did not [2, 14, 19, 30] Kahalon et al. [24] found trait self-objectification moderated the effect of condition (i.e., receiving an appearance compliment) on math test scores, but did not replicate this finding in the paper's second study. Based on the means and standard deviations reported in the second study, women in the experimental condition performed worse on a math test than those in the control condition by a Cohen's $d$ of 0.52 (a moderate-to-large effect), *without* accounting for trait self-objectification or previous math scores. Of the two studies that found evidence of trait self-objectification or a similar trait-like variable as a moderator [31, 32], neither found a significant main effect for their objectification manipulation on performance. In summary, some previous studies found that objectification manipulations increase self-objectification and impair cognitive performance regardless of trait self-objectification. Other studies have only been able to show these effects when using trait self-objectification as a moderator. In other words, based on empirical findings, it is not clear that researchers must control for trait-level self-objectification to see the effects of objectification manipulations. However, given the relevance of this question to theory development, we recommend that future researchers obtain a measure of trait self-objectification if it is possible to do so without sensitizing participants to a study's purpose or hypotheses.

There are strong theoretical reasons to predict that objectification would have a negative effect on women's thought processes and cognitive performance. It is well established that attention is a limited resource (for a review, see Scalf et al. [78]) and that negative, self-focused attention can be particularly disruptive to cognitive processes [79]. It follows that self-objectification, a form of appearance-focused self-referential thinking, could usurp a portion of women's conscious attention as originally proposed by Fredrickson and Roberts [1]. However, our study found no evidence to support this proposal, despite employing a task that required significant attentional resources. Eronen and Bringmann [80] argue for a bidirectional relationship between theory and phenomena, such that theories predict observable phenomena and phenomena impose constraints on possible theories. They argue phenomena should be "verifiable and detectable in several independent ways and not dependent on theoretical framework or observation method" (p. 780). It follows that null results such as those found in the current study are essential to our understanding of this phenomenon's robustness and provide possible constraints on theoretical assumptions regarding objectification's impact on women's cognitive performance.

## Implications and future directions

The present study is distinct from previous work on this topic due to our focus on statistical power and preregistration. Scholars have recently argued that a large proportion of the findings in psychology literature are "false positives" resulting from underpowered studies and a bias toward reporting only statistically significant findings [21, 27]. Underpowered studies are harmful for two main reasons: they decrease the likelihood of detecting a true effect and

increase the likelihood of reporting inflated effect sizes if the result is statistically significant. Given many of the studies on objectification and cognitive performance were likely underpowered for key analyses, the expected effect size of the impact of objectification on women's performance remains unclear. Moreover, researchers are recognizing that flexibility in data collection, methods, and analysis also increases the chances of producing false-positive findings [38]. To combat this issue, we conducted an a priori power analysis to determine the minimum sample size needed and adhered to the methods and analysis plan outlined in our preregistration.

Though null results from a single study should never be interpreted as overturning an existing body of literature, the preregistration and high level of statistical power for the current study suggest these results should at least leave researchers open to considering the extent to which findings concerning objectification's impact on women's cognitive performance are broad or replicable. It is possible that the effect size of objectification's influence on women's performance is inflated in the current literature and that the effect was too small to detect even with the large sample we employed. However, recognizing that published effect sizes may be inflated [27], we reduced our expected effect size from a Cohen's $d$ of 0.61 (found in Fredrickson et al. [2]) to a more modest 0.35.

In the future, similarly well-powered studies should continue to investigate whether there is an effect of objectifying environments on women's cognitive performance and if so, which types of cognitive tasks are affected. Self-objectification may disrupt certain types of cognitive processes more than others. The dependent variables employed in the current literature include a wide array of cognitive tests and tasks. It seems likely that the choice of dependent variable could have a large impact on the ability to detect effects of objectification on women's cognitive performance. Future work should focus on selecting dependent variables that are theoretically driven (i.e., cognitive tasks likely to be disrupted by body monitoring) and are sensitive to even minor increases in working memory load. As noted above, the ability to control for individual-level differences likely to affect performance would also be ideal.

Since the current study was conducted, the need for realistic and robust research in this area has only grown. Preliminary evidence suggests links between appearance concern and fatigue due to video conferencing [81]. Additional recent work suggests that appearance distraction and self-focused attention may mediate the effect of appearance concerns on perceived video call performance [82]. The current study provides strong evidence for an increase in women's state self-objectification when video recorded versus audio-recorded only. However, the current study focused on video recording from the neck down as a simulation of objectification, rather than recording the head and shoulders which is more similar to the way one is framed while video conferencing. One recent study found that, when video conferencing, women on-camera (without self-view) experienced reduced cognitive performance compared to those off-camera [30]. However, the authors found no evidence that increases of state self-objectification explained this relationship. Further, women with both their camera and self-view on did not experience reduced performance. Given the persistence of remote work and schooling, future work should continue to examine the potential burdens women face while being on camera, including the possible impact on cognitive performance [30]. Undoubtedly, women experience many high-stakes situations where both their appearance and performance are simultaneously evaluated (e.g., job interviews) on- and offline. It remains unclear how an emphasis on a woman's appearance might interact with her ability to perform in these settings.

Work in this area thus far has focused on how relatively small objectification manipulations can have large, negative impacts on women. However, publication bias for statistically significant findings and the tendency of underpowered studies to produce inflated effect sizes could

suggest that the effect of objectification's impact on women's cognitive performance may be overstated. Theory and empirical evidence alike suggest that objectification is an everyday experience for women [4, 12, 72], making reliable conclusions about objectification a concern not only for academics but also a practical concern for women's well-being. The null results of this study could be an indication of women's resilience in the face of objectification, a premise left largely unexplored in the current literature. Future research should investigate women's ability to combat objectifying environments and self-objectifying thoughts, reclaiming their minds from the world's focus on their bodies.

## Acknowledgments

Thank you to Northwestern's Body and Media Lab for assistance with data collection.

## Author Contributions

**Conceptualization:** Anne Zola, Renee Engeln.

**Data curation:** Anne Zola, Renee Engeln.

**Formal analysis:** Anne Zola, Renee Engeln.

**Funding acquisition:** Anne Zola, Renee Engeln.

**Investigation:** Anne Zola, Renee Engeln.

**Methodology:** Anne Zola, Renee Engeln.

**Project administration:** Anne Zola, Renee Engeln.

**Supervision:** Renee Engeln.

**Writing – original draft:** Anne Zola, Renee Engeln.

**Writing – review & editing:** Anne Zola, Renee Engeln.

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
