## [Decision Letter · Decision Letter 0]

29 Nov 2022

PONE-D-21-22853Brains Over Beauty: A Preregistered Test of the Effects of Objectification on Women’s Cognitive PerformancePLOS ONE

Dear Dr. Zola,

thank you for submitting your manuscript to PLOS ONE.  After careful consideration by several experts in the field with (you will see) somewhat disparate opinions, we feel that it has merit but as it currently stands, has to be improved in order to fully meet PLOS ONE’s publication criteria.  Therefore, we invite you to submit a revised version of the manuscript that addresses the points raised during the review process.

Please make sure to carefully, point-by-point respond to the concerns raised by Reviewers, paying particular attention to the technical aspects of your work.  Also, pease explain and discuss any differences of your research from the original one by Gay and Castano as mentioned by Reviewer 1.

Please submit your revised manuscript within six months from this date as thereafter, any revision has to be considered a new submission.  If you will need more time than this to complete your revisions, please reply to this message or contact the journal office at plosone@plos.org.  Please include the following items when submitting your revised manuscript:A rebuttal letter that responds to each point raised by the academic editor and reviewer(s). You should upload this letter as a separate file labeled 'Response to Reviewers'.A marked-up copy of your manuscript that highlights changes made to the original version. You should upload this as a separate file labeled 'Revised Manuscript with Track Changes'.An unmarked version of your revised paper without tracked changes. You should upload this as a separate file labeled 'Manuscript'.

We look forward to receiving your revised manuscript.  Thank you for choosing PLOS ONE for reporting your research.

Kind regards,

Sasha

Alexander N. 'Sasha' Sokolov, Ph.D.

Academic Editor

PLOS ONE

Journal Requirements:

“This research was supported by grants from the Northwestern University Undergraduate Research Grants Program (awarded to AZ).The funders had no role in study design, data collection and analysis, decision to publish, or preparation of the manuscript.”

3**. **Please include your full ethics statement in the ‘Methods’ section of your manuscript file. In your statement, please include the full name of the IRB or ethics committee who approved or waived your study, as well as whether or not you obtained informed written or verbal consent. If consent was waived for your study, please include this information in your statement as well.

Reviewers' comments:

Reviewer's Responses to Questions

**Comments to the Author**

1. Is the manuscript technically sound, and do the data support the conclusions?

Reviewer #1: No

Reviewer #2: Yes

Reviewer #3: Yes

2. Has the statistical analysis been performed appropriately and rigorously? 

Reviewer #1: Yes

Reviewer #2: Yes

Reviewer #3: Yes

3. Have the authors made all data underlying the findings in their manuscript fully available?

Reviewer #1: Yes

Reviewer #2: Yes

Reviewer #3: Yes

4. Is the manuscript presented in an intelligible fashion and written in standard English?

Reviewer #1: Yes

Reviewer #2: Yes

Reviewer #3: Yes

5. Review Comments to the Author

Reviewer #1: The authors correctly point to the fact that the experiments testing the effects of objectification have been typically underpowered, and that this may account for some inconsistent results. They also suggest that in these studies the use of covariates and of moderating variables might have been too ad-hoc, and "after the fact." Considering that the authors have no evidence of any of this, I found their statement in this regard (p. 11) unwarranted. This is a small detail, however.

To correct all this, they set out to conduct a well-powered, pre-registered experiment, with a three-level condition intended to test the effect of male gaze on cognitive performance of women.  I truly applaud their systematic approach, the effort, and the fact that they pre-registered the study and shared all the data. The experiment, however, suffers from three major methodological flaws that make the results unpublishable.

1. Previous research, discussed by these authors, finds that the effects of an objectification manipulation is likely moderated by trait self-objectification. Objectification scholars clearly theorize this, as the authors themselves write on page 12 "As noted, self-objectification can be conceptualized as both a state and trait variable (Noll & Fredrickson, 1998), and therefore, in line with original theory, women who have the tendency to self-objectify could have a greater chance of being distracted by body monitoring."  Yet, the authors decide NOT to take any trait measure of self-objectification. This alone could be the reason why they did not find effects of their manipulation (moderated by TSO, of course).

2. The authors seem to model their manipulation after the study by Gay and Castano, in which the above-mentioned interaction was found. Yet, instead of having women in the male-gaze condition filmed by a man, they only tell them that the video might be later viewed by a man. Why the authors chose to water-down so much a manipulation that not only had worked in the past, but that it is also a direct operazionalization of the theory of objectification and it has excellent ecological validity, is puzzling to me.

3. Previous research also shows that the task must be somewhat challenging for the effects of objectification to emerge. This is true, by the way, also for other social cognition research that does not deal with objectification: If it is too easy, you won't find the effect of the manipulation because participants can complete the task even when they are cognitively depleted. Yet, here too, the authors make the puzzling choice of selecting 20 very easy items: range 0-20, M = 16.33.

Reviewer #2: Thank you for the opportunity to review, “Brains over beauty: A pre-registered test of the effects of objectification on women’s cognitive performance.” The authors should be commended for conducting work that is pre-registered and adds clarity not only to an important tenant of objectification theory, but also to the literature that has provided mixed results in terms of the relation between women’s self-objectification and cognitive performance. I also would like to say that I thought this manuscript was very well-written and easy to follow, from the literature review to coverage of the results. Below I outline a few points that I believe could make this an even stronger manuscript.

I appreciated that the authors took time to outline all previous work conducted on self-objectification and cognitive performance. I found myself trying to trace citations to determine whether work that found significant effects was one in the same with work that was potentially underpowered. This made me think that including a table of all papers, with columns of all relevant pieces of information (e.g., manipulation, sample size, findings), could make for an easier examination of inconsistencies in past findings and possible reasons why (e.g., which seven of the studies were likely underpowered?).

Given that the Guizzo paper assessed the closest concept to the current work - sustained attention - can the authors elaborate on what this task looked like? Could you also clarify that Guizzo’s moderator of internalized beauty ideals was an conceptualization of trait level self-objectification?

My only true concern with this manuscript is that it seems the authors may have missed some key citations. For instance, Winn & Cornelius have a 2020 review of literature on the topic, which I think if referenced would provide further justification for the previous mixed results and need for clarity. Additionally, a few other studies came to my mind that examine the relation between self-objectification and cognitive performance (broadly defined): Baldissari & Andrighetto (2021), Aubrey & Gerding (2015), and the sexual harassment program of work from Gervais & Wiener. While the authors may have purposely not included these articles in their literature review, it’d be helpful to understand how they came to the conclusion that there are only 9 studies on the topic.

In the intro, the authors mention that much previous work includes covariates in analyses. While true, I think it is an inappropriate assumption that this is a post-hoc decision. It’s true we cannot know without pre-registration, but many of the covariates are supported by research or theory, so I feel like this language should be tempered. In the authors’ current stance, it seems as if they think this is always an unacceptable practice, but I’m not sure I agree.

Fredrickson & Roberts suggest that not all women will respond to instances of objectification in the same manner. This is also most likely true of experiences that prompt self-objectification, whether intra or interpersonally - for women high in trait self-objectification, they are likely to engage in higher levels of self-objectification than those lower in this trait. As a result, I was surprised when the authors revealed the correlations between state body surveillance and performance without taking condition into consideration. Because women in the control condition did not experience any environmental prompt to self-objectify, body surveillance measured among these participants would likely be akin to a trait level of self-objectification. Although not pre-registered, I would like to see these additional analyses at least in a footnote.

One issue I have with how researchers discuss manipulations involving objectification is a lack of specificity. While many claim to manipulate self-objectification, they are actually manipulating interpersonal objectification, from which increased self-objectification levels ensue. I think when discussing the manipulations used, this paper (and the literature more broadly) may benefit from a bit more specificity here in terms of what is truly being manipulated.

Within the discussion, I’d like to see more take-away points from the current work. Specifically, while the authors contrast their findings with those of past work, how do these findings inform objectification theory? Moreover, beyond calls for replication and power analyses, do these findings have implications for how objectification researchers conduct their work (e.g., in a literature with such varied ways of manipulating objectification, are the two manipulations in the current work interchangeable?)?

Smaller details:

I felt the abstract could have used a bit more specificity as well as a bit more elaboration about the implication of the findings.

While the authors note that the sample in the current work may have differed from past samples in terms of math abilities, the current sample seems demographically similar to past samples in terms of age and race. I think this similarity is worth mentioning. I also think it’s worth mentioning in the introduction that affect was included because of previous findings.

Because the authors critically analyzed sample sizes of past work, could you benefit future work by providing the number of participants per condition after exclusions?

I hope this feedback is helpful to the authors and I look forward to seeing this paper in its final state!

Reviewer #3: The paper “Brains over Beauty” tested, using a more rigorous methodology than that used in previous research, whether the induction of state self-objectification interferes with women’s cognitive performance. The paper has several advantages over past research in that it (a) used a sufficiently powered sample, (b) examined two different types of manipulations to induce self-objectification, and (c) used a dependent variable that is unlikely to be influenced by stereotype threat (which could be an alternative explanation to the effect(s) observed in several of the previous studies). Because of these strengths, and because finding out what *does* not work despite being theoretically plausible is highly important for scientific advancement (see Eronen & Bringmann, 2021), I think that the paper makes an important contribution to the literature on women’s sexual objectification. Nevertheless, I identified several weaknesses, which I list below.

1. In my reading of objectification theory, the original 1997 paper by Fredrickson and Roberts aimed to explain (through the concept of self-objectification) why women experience higher rates of unipolar depression, eating disorders, and sexual dysfunctions as compared to men. The idea that self-objectification should interfere with cognitive performance, tested in the 1998 paper by Fredrickson et al., is an extension of the original theory (as put forward in the 1997 paper). I think that the literature review in the present paper should reflect this (unless the authors have a different view on how objectification theory evolved, and if so – perhaps they can explain their view, at least in the response letter).

2. When you discuss the research by Quinn et al. (2006) please note that there is a major flaw in how the DV (performance in a Stroop test) was calculated in this study (instead of looking at participants’ interference score, the authors looked at the overall reaction times without comparing congruent vs. incongruent trials – which is NOT how performance in a Stroop test should be calculated).

3. I think that it will be very helpful to the readers if you could add a table summarizing the main findings of experimental research on the effects of state self-objectification (SSO) on cognitive performance. That is, for each research mention (in a separate column) how was SSO manipulated? what was the DV? what was the main finding(s)? was the study sufficiently powered? This information appears in the text, but it will be much more convenient for readers to have it all briefly concentrated in one place (in addition to the longer description that currently appears in the text).

4. The term “stereotype threat” is mentioned but not defined. I suggest briefly explaining what it means (as you cannot assume that all readers are familiar with this literature).

5. To the best of my knowledge, it is recommended to quantify the evidence in favor of the null hypothesis (which seems to be supported in the present study) using Bayesian hypothesis testing (e.g., Wagenmakers et al., 2018).

6. In p. 18 there is a link to the test that participants had to solve, but it took me some time to find in which document it is located (because the link brings the reader to a list of 9 documents). So, you can say something like “see here (in the document titled MATH appendixes) for list of problems”.

7. I wonder whether the manipulation in the “male gaze” condition could be strengthened by leading participants to believe that they are going to meet with a man who saw their video (taped from the neck down).

8. I think that the fact that the authors used a simple (rather than a difficult) math task is an advantage of the present study. In my understanding, there is no theoretical basis to the prediction that the effects of SSO should appear for difficult rather than simple tasks. In other words, I don’t agree with the authors’ suggestion, in p. 27 in the GD, that “objectification may have a greater impact on difficult compared to easy tasks,” and that this possibility should be tested in future research. The use of a simple math task, as done by the authors, is correct. However, I do think that a measure of participants’ pre-existing math ability/performance or at least their math identification should have been included and controlled for. I understand that the random assignment to experimental conditions should “take care” of any pre-existing differences in math ability. Nevertheless, the results would be more convincing if the authors could show that they persist even when controlling for pre-existing differences in math performance and/or identification. This is because there are huge interpersonal differences in these variables, which can create a lot of “statistical noise”.

9. In p. 28, you refer to the Fredrickson et al.’s (1998) study as if participants were “completing an advanced math test in a room alone while wearing a bathing suit”. To the best of my memory, participants in this study first tried on a bathing suit in front of a mirror, but then completed the math test wearing their regular clothes. Please double check it to verify that the information is correct.

10. In the GD, you discuss trait self-objectification (TSO) as a potential moderator. I think that not testing for moderation by TSO is a major limitation of the present study. The authors explain the choice not to measure TSO prior to the experimental manipulation in that they didn’t want to prime this concept. I agree with this explanation, but I think that at the very least they should mention that future studies can overcome this problem by splitting the study into two parts, such that in the first part the potential moderators (including TSO) are measured, and in the second part (which can be held one week later, or so) the experimental manipulation is induced and the DVs are measured (as done by Kahalon et al., 2018).

11. Another potential moderator that can be proposed in the GD (to be tested in the future) is women’s enjoyment of their sexualization (e.g., Liss et al., 2011). It makes sense the negative effects on mood are observed for women who are low on this measure, but not for women who enjoy being admired by men.

6. PLOS authors have the option to publish the peer review history of their article (what does this mean?). If published, this will include your full peer review and any attached files.

Reviewer #1: No

Reviewer #2: No

Reviewer #3: No

---

## [Author Response · Author response to Decision Letter 0]

22 Mar 2023

Response to Editor and Reviewer comments

Please explain and discuss any differences of your research from the original one by Gay and Castano as mentioned by Reviewer 1.

We have added additional details in both the introduction and discussion section on the differences between the current study and other studies where participants were videoed (Gay & Castano, 2010) or photographed (Guizzo & Cadinu, 2017) by a male or female experimenter. 

“This research was supported by grants from the Northwestern University Undergraduate Research Grants Program (awarded to AZ). The funders had no role in study design, data collection and analysis, decision to publish, or preparation of the manuscript.”

We have included an amended funding statement in the edited manuscript and in the response letter above. 

We have now included a full ethics statement in the Methods section, with the full name of the IRB and IRB number. We have clarified that participants provided written consent in order to participate.

Reviewer #1: The authors correctly point to the fact that the experiments testing the effects of objectification have been typically underpowered, and that this may account for some inconsistent results. They also suggest that in these studies the use of covariates and of moderating variables might have been too ad-hoc, and "after the fact." Considering that the authors have no evidence of any of this, I found their statement in this regard (p. 11) unwarranted. This is a small detail, however.

We have removed this statement about ad-hoc hypotheses as requested and have tempered our language on this issue throughout. 

To correct all this, they set out to conduct a well-powered, pre-registered experiment, with a three-level condition intended to test the effect of male gaze on cognitive performance of women. I truly applaud their systematic approach, the effort, and the fact that they pre-registered the study and shared all the data. The experiment, however, suffers from three major methodological flaws that make the results unpublishable.

1. Previous research, discussed by these authors, finds that the effects of an objectification manipulation is likely moderated by trait self-objectification. Objectification scholars clearly theorize this, as the authors themselves write on page 12 "As noted, self-objectification can be conceptualized as both a state and trait variable (Noll & Fredrickson, 1998), and therefore, in line with original theory, women who have the tendency to self-objectify could have a greater chance of being distracted by body monitoring." Yet, the authors decide NOT to take any trait measure of self-objectification. This alone could be the reason why they did not find effects of their manipulation (moderated by TSO, of course).

The reviewer raises the concern that previous research on this topic shows the effect of state self-objectification on conscious attention is moderated by trait self-objectification (TSO). Although moderators can certainly be important to consider when testing theoretically derived hypotheses, our reading of the published literature on this topic does not lead us to conclude that TSO is an essential moderator when it comes to the effect of objectification on cognitive performance. For example, despite not including TSO as a moderator, Quinn et al. (2006) reported a significant main effect of objectification on a modified Stroop task performance and Gervais (2011) found women in the experimental condition performed worse on a timed math test relative to the control. Of those studies that do measure TSO or a similar trait-level variable, the results are mixed. Four studies (Tiggemann & Boundy, 2008; Hebl et al., 2004; Gapinski et al., 2003; Fredrickson et al., 1998) found no evidence that trait self-objectification moderates the effect of objectification on performance. Of these, one reports a significant main effect of objectification on performance with a cut-off p < .05(Hebl et al., 2004) and three do not (Tiggemann & Boundy, 2008; Gapinski et al., 2003; Fredrickson et al., 1998). In Study 1, Kahalon et al. (2018) found trait self-objectification moderated the effect of condition (i.e., receiving an appearance compliment) on math test scores, but did not replicate this finding in Study 2. Based on the means and standard deviations reported in Study 2, women in the experimental condition performed worse on a math test than those in the control condition by a Cohen’s d of 0.52 (a moderate-to-large effect), without accounting for TSO or previous math scores. 

We noted two studies that found evidence of TSO or a similar trait-like variable as a potential moderator. Guizzo and Cadinu (2017) found no effect of condition on sustained attention to response task but reported internalized beauty ideals (a trait-level variable with similarities to TSO) as a significant moderator of this relationship. Gay and Castano (2010) used a manipulation that involved either a male or female experimenter video recording participant as they walked down a hallway, but did not include a no-video control in their study. They did not find a main effect of experimental condition (male vs. female experimenter) on a letter-number sequencing (LNS) task, LNS latency, or GMAT scores in Study 1 (n = 25) or Study 2 (n = 50). In Study 1, they did find a TSO by condition interaction on LNS latency, whereby participants with higher TSO performed worse in the condition with the male experimenter compared to the condition with the female experimenter (p = .02). The same TSO by condition interaction did not reach statistical significance in Study 2 (p = .07). 

Overall, findings on TSO as a moderator of objectification’s impact on cognitive performance are mixed. We now address these findings more specifically in the discussion section, to help situate our findings in the current body of literature on this topic.

We designed our experiment to have sufficient power for the primary effect of interest (one-way ANOVA with 3 conditions) and found a moderate effect of experimental condition on body surveillance (i.e., state self-objectification). Although it is possible that participants with high TSO would be even more likely to experience state self-objectification (and thus be at greater risk of distraction during the task), we reasoned that participants who experience higher levels of state self-objectification would perform worse on the working memory task, even in the control condition. However, we found no significant associations between body surveillance and performance or latency on the working memory task for any condition. Nonetheless, we agree with the reviewer that it would be ideal if we had participants’ TSO scores for this study and address this as a limitation in our discussion section. 

2. The authors seem to model their manipulation after the study by Gay and Castano, in which the above-mentioned interaction was found. Yet, instead of having women in the male-gaze condition filmed by a man, they only tell them that the video might be later viewed by a man. Why the authors chose to water-down so much a manipulation that not only had worked in the past, but that it is also a direct operationalization of the theory of objectification and it has excellent ecological validity, is puzzling to me.

We agree with the reviewer that the video recording manipulation used in Gay and Castano has excellent ecological validity and therefore, modeled our experiment on the premise that framing a woman’s body with a video camera from the neck down would simulate an objectifying gaze. However, instead of having a male vs. female experimenter video record participants, we opted to use an anticipated male gaze manipulation as in Calogero et al. (2004), as Calogero et al. found that anticipating a male gaze increased social physique anxiety (a construct that overlaps with body surveillance). Across two studies, Gay and Castano did not include a manipulation check (i.e., a measure of body surveillance/state self-objectification), so it is difficult to evaluate the extent to which their precise manipulation worked – especially since they found no main effects of their male vs. female experimenter manipulation. Gay and Castano did report some statistically significant effects for tests of two-way interactions and a three-way interaction (with TSO and item difficulty), but one interaction suggested that the female experimenter (vs. the male experimenter) had a more negative effect on latency. Given the very low sample sizes in Gay and Castano (25 participants in one study and 50 in another), it is difficult to know what to make of these interactions and the likelihood of false positive findings in these analyses appears high. Finally, unlike other studies that have used framing women’s bodies with a camera (Gay & Castano, 2010; Guizzo & Cadinu, 2017), we included a third condition with no video camera as a control, which we believe allows for a more direct test of our key hypothesis than using male vs. female experimenters as a manipulation, especially given evidence that female experimenters can also trigger self-objectification in women (Tiggemann & Boundy, 2008). 

In our discussion section, we acknowledge that despite its efficacy in Calogero et al. (2004), it is possible that our anticipated male gaze manipulation was not strong enough and that this could explain why we did not see significantly different levels of body surveillance between our two experimental conditions (and found significant differences only between our experimental conditions and our control). Nonetheless, we observed no effect of condition on performance despite a moderate effect of the control versus the experimental conditions on body surveillance; we focus on results between experimental conditions and control rather than differences between the two experimental conditions. 

3. Previous research also shows that the task must be somewhat challenging for the effects of objectification to emerge. This is true, by the way, also for other social cognition research that does not deal with objectification: If it is too easy, you won't find the effect of the manipulation because participants can complete the task even when they are cognitively depleted. Yet, here too, the authors make the puzzling choice of selecting 20 very easy items: range 0-20, M = 16.33.

We agree with the reviewer that if a task is too easy, participants may be able to successfully complete it even while cognitively depleted. In the case of our DV, the items required only basic arithmetic skills, but were cognitively taxing to solve because they had to be solved without pen and paper (i.e., in one’s head). In fact, solving mental math problems is commonly used in cognitive psychology as a manipulation that loads working memory (e.g., Ashcraft & Krause, 2007). We now include more discussion of the working memory task we used in the paper and provide citations to support the contention that these mental math problems were likely taxing to participants even if they were able to generate the correct answers. As evidence of this conclusion, the latency scores for these math problems had a substantial amount of variability. 

Reviewer #2: Thank you for the opportunity to review, “Brains over beauty: A pre-registered test of the effects of objectification on women’s cognitive performance.” The authors should be commended for conducting work that is pre-registered and adds clarity not only to an important tenant of objectification theory, but also to the literature that has provided mixed results in terms of the relation between women’s self-objectification and cognitive performance. I also would like to say that I thought this manuscript was very well-written and easy to follow, from the literature review to coverage of the results. Below I outline a few points that I believe could make this an even stronger manuscript.

I appreciated that the authors took time to outline all previous work conducted on self-objectification and cognitive performance. I found myself trying to trace citations to determine whether work that found significant effects was one in the same with work that was potentially underpowered. This made me think that including a table of all papers, with columns of all relevant pieces of information (e.g., manipulation, sample size, findings), could make for an easier examination of inconsistencies in past findings and possible reasons why (e.g., which seven of the studies were likely underpowered?).

Thank you for this suggestion. We have now included a table that outlines the manipulation, dependent variable, and primary outcomes of previous work in this area. 

Given that the Guizzo paper assessed the closest concept to the current work - sustained attention - can the authors elaborate on what this task looked like? Could you also clarify that Guizzo’s moderator of internalized beauty ideals was an conceptualization of trait level self-objectification?

Additional details about this paper are now included (see p. 9). 

My only true concern with this manuscript is that it seems the authors may have missed some key citations. For instance, Winn & Cornelius have a 2020 review of literature on the topic, which I think if referenced would provide further justification for the previous mixed results and need for clarity. Additionally, a few other studies came to my mind that examine the relation between self-objectification and cognitive performance (broadly defined): Baldissari & Andrighetto (2021), Aubrey & Gerding (2015), and the sexual harassment program of work from Gervais & Wiener. While the authors may have purposely not included these articles in their literature review, it’d be helpful to understand how they came to the conclusion that there are only 9 studies on the topic.

We appreciate these suggestions for additional literature to include. We carefully considered the studies cited in the Winn and Cornelius (2020) review and have incorporated additional relevant studies from that review (including Aubrey & Gerding, 2015). We have not included Baldissari & Andrighetto (2021), as that paper examined a non-sexualized form of objectification that is less relevant to body monitoring. In general, we did not include literature focused on sexual harassment, consistent with the distinction in objectification theory research between body evaluation and unwanted explicit sexual advances (e.g., Kozee et. al., 2007). 

In the intro, the authors mention that much previous work includes covariates in analyses. While true, I think it is an inappropriate assumption that this is a post-hoc decision. It’s true we cannot know without pre-registration, but many of the covariates are supported by research or theory, so I feel like this language should be tempered. In the authors’ current stance, it seems as if they think this is always an unacceptable practice, but I’m not sure I agree.

Thank you for this suggestion. We have removed this statement and have tempered our language throughout. 

Fredrickson & Roberts suggest that not all women will respond to instances of objectification in the same manner. This is also most likely true of experiences that prompt self-objectification, whether intra or interpersonally - for women high in trait self-objectification, they are likely to engage in higher levels of self-objectification than those lower in this trait. As a result, I was surprised when the authors revealed the correlations between state body surveillance and performance without taking condition into consideration. Because women in the control condition did not experience any environmental prompt to self-objectify, body surveillance measured among these participants would likely be akin to a trait level of self-objectification. Although not pre-registered, I would like to see these additional analyses at least in a footnote.

As requested, we have added correlations separated by condition between body surveillance, response accuracy, and response speed to the results section. 

One issue I have with how researchers discuss manipulations involving objectification is a lack of specificity. While many claim to manipulate self-objectification, they are actually manipulating interpersonal objectification, from which increased self-objectification levels ensue. I think when discussing the manipulations used, this paper (and the literature more broadly) may benefit from a bit more specificity here in terms of what is truly being manipulated.

We have added notes in the table of previous studies indicating whether the manipulation was focused on manipulating state self-objectification or focused on manipulated interpersonal objectification, which can result in increased state self-objectification. We hope this indicator and the description of the manipulation in the table can help clarify which manipulations were used in previous studies.

Within the discussion, I’d like to see more take-away points from the current work. Specifically, while the authors contrast their findings with those of past work, how do these findings inform objectification theory? Moreover, beyond calls for replication and power analyses, do these findings have implications for how objectification researchers conduct their work (e.g., in a literature with such varied ways of manipulating objectification, are the two manipulations in the current work interchangeable?)?

We thank the reviewer for this suggestion. We have substantially reworked the discussion sections to make the takeaway points clearer. We now discuss how our work contrasts with previous work in terms of objectification manipulation and working memory task. We urge future researchers to consider the differences between interpersonal and non-interpersonal objectification manipulations and highlight the theoretical reasons we think these manipulations are not interchangeable. We also include additional discussion regarding potential pathways for future research in this area, including women’s potential cognitive burden while being video conferencing. 

Smaller details:

I felt the abstract could have used a bit more specificity as well as a bit more elaboration about the implication of the findings.

We have included additional details in the abstract and discussed the implications of the range of manipulations and dependent variables used in this area of research.

While the authors note that the sample in the current work may have differed from past samples in terms of math abilities, the current sample seems demographically similar to past samples in terms of age and race. I think this similarity is worth mentioning. I also think it’s worth mentioning in the introduction that affect was included because of previous findings.

Thank you for these suggestions. We now clarify in the discussion section that our sample was similar in terms of age and race as used in previous studies on this topic. In the Introduction, we clarify that affect outcomes (affect and guilt/shame) were included based on findings from previous work.

Because the authors critically analyzed sample sizes of past work, could you benefit future work by providing the number of participants per condition after exclusions?

Thank you for this suggestion. We have added sample sizes per condition after exclusions to the text and to the tables.

I hope this feedback is helpful to the authors and I look forward to seeing this paper in its final state!

Reviewer #3: The paper “Brains over Beauty” tested, using a more rigorous methodology than that used in previous research, whether the induction of state self-objectification interferes with women’s cognitive performance. The paper has several advantages over past research in that it (a) used a sufficiently powered sample, (b) examined two different types of manipulations to induce self-objectification, and (c) used a dependent variable that is unlikely to be influenced by stereotype threat (which could be an alternative explanation to the effect(s) observed in several of the previous studies). Because of these strengths, and because finding out what *does* not work despite being theoretically plausible is highly important for scientific advancement (see Eronen & Bringmann, 2021), I think that the paper makes an important contribution to the literature on women’s sexual objectification. Nevertheless, I identified several weaknesses, which I list below.

We appreciate the reviewer pointing to recent discussions about the strengths of null results and now incorporate arguments made by Eronen and Bringmann (2021) in our discussion.

1. In my reading of objectification theory, the original 1997 paper by Fredrickson and Roberts aimed to explain (through the concept of self-objectification) why women experience higher rates of unipolar depression, eating disorders, and sexual dysfunctions as compared to men. The idea that self-objectification should interfere with cognitive performance, tested in the 1998 paper by Fredrickson et al., is an extension of the original theory (as put forward in the 1997 paper). I think that the literature review in the present paper should reflect this (unless the authors have a different view on how objectification theory evolved, and if so – perhaps they can explain their view, at least in the response letter).

Our perspective is that the original 1997 paper laid the foundation for the claim that self-objectification could interfere with cognitive performance, arguing that "significant portions of women's conscious attention can often be usurped by concerns related to real or imagined, present or anticipated, surveyors of their physical appearance." The original paper also mentioned these types of cognitive effects when discussing how self-objectification can inhibit women’s movement: “These fits and starts apparent in women's movements may also affect women's mental concentration.” However, the reviewer makes a good point that this idea was more directly elaborated upon in Fredrickson et al. (1998). We have now updated the wording in the literature review to more accurately reflect this. 

2. When you discuss the research by Quinn et al. (2006) please note that there is a major flaw in how the DV (performance in a Stroop test) was calculated in this study (instead of looking at participants’ interference score, the authors looked at the overall reaction times without comparing congruent vs. incongruent trials – which is NOT how performance in a Stroop test should be calculated).

We thank the reviewer for their careful reading of the literature and have clarified in the manuscript that Quinn et al. (2006) employed a modified Stroop task.

3. I think that it will be very helpful to the readers if you could add a table summarizing the main findings of experimental research on the effects of state self-objectification (SSO) on cognitive performance. That is, for each research mention (in a separate column) how was SSO manipulated? what was the DV? what was the main finding(s)? was the study sufficiently powered? This information appears in the text, but it will be much more convenient for readers to have it all briefly concentrated in one place (in addition to the longer description that currently appears in the text).

Thank you for this suggestion. We have now included a table that outlines the manipulation, dependent variable, and primary outcomes of previous work. 

4. The term “stereotype threat” is mentioned but not defined. I suggest briefly explaining what it means (as you cannot assume that all readers are familiar with this literature).

We now provide a brief definition of stereotype threat in the manuscript.

5. To the best of my knowledge, it is recommended to quantify the evidence in favor of the null hypothesis (which seems to be supported in the present study) using Bayesian hypothesis testing (e.g., Wagenmakers et al., 2018).

Thank you for the suggestion and resource. We now include calculated Bayes factors in the text and in the supplementary materials online.

6. In p. 18 there is a link to the test that participants had to solve, but it took me some time to find in which document it is located (because the link brings the reader to a list of 9 documents). So, you can say something like “see here (in the document titled MATH appendixes) for list of problems”.

We have added this clarification to the manuscript. 

7. I wonder whether the manipulation in the “male gaze” condition could be strengthened by leading participants to believe that they are going to meet with a man who saw their video (taped from the neck down).

We now include additional discussion regarding ways the male gaze condition could have been strengthened, including whether the woman thought she would interact with the man who viewed her video.

8. I think that the fact that the authors used a simple (rather than a difficult) math task is an advantage of the present study. In my understanding, there is no theoretical basis to the prediction that the effects of SSO should appear for difficult rather than simple tasks. In other words, I don’t agree with the authors’ suggestion, in p. 27 in the GD, that “objectification may have a greater impact on difficult compared to easy tasks,” and that this possibility should be tested in future research. The use of a simple math task, as done by the authors, is correct. However, I do think that a measure of participants’ pre-existing math ability/performance or at least their math identification should have been included and controlled for. I understand that the random assignment to experimental conditions should “take care” of any pre-existing differences in math ability. Nevertheless, the results would be more convincing if the authors could show that they persist even when controlling for pre-existing differences in math performance and/or identification. This is because there are huge interpersonal differences in these variables, which can create a lot of “statistical noise”.

We have added text to the discussion about the limitations of not controlling for previous math scores or math identification.

9. In p. 28, you refer to the Fredrickson et al.’s (1998) study as if participants were “completing an advanced math test in a room alone while wearing a bathing suit”. To the best of my memory, participants in this study first tried on a bathing suit in front of a mirror, but then completed the math test wearing their regular clothes. Please double check it to verify that the information is correct.

Thank you for your close reading of the manuscript and the suggestion to verify this information. We have carefully reviewed Fredrickson et al.’s (1998) study as well as the other studies that use the swimsuit vs. sweater manipulation. We can confirm they report in their procedure section that participants wore the swimsuit or sweater while completing the math test and additional outcome measures.

10. In the GD, you discuss trait self-objectification (TSO) as a potential moderator. I think that not testing for moderation by TSO is a major limitation of the present study. The authors explain the choice not to measure TSO prior to the experimental manipulation in that they didn’t want to prime this concept. I agree with this explanation, but I think that at the very least they should mention that future studies can overcome this problem by splitting the study into two parts, such that in the first part the potential moderators (including TSO) are measured, and in the second part (which can be held one week later, or so) the experimental manipulation is induced and the DVs are measured (as done by Kahalon et al., 2018).

We have included additional discussion about the role of TSO as a moderator in previous work and our reasoning for not including it as a moderator in the current study. We have added the suggestion that future research could separate the study into two parts to avoid demand characteristics when measuring TSO or a similar trait-like variable. 

11. Another potential moderator that can be proposed in the GD (to be tested in the future) is women’s enjoyment of their sexualization (e.g., Liss et al., 2011). It makes sense the negative effects on mood are observed for women who are low on this measure, but not for women who enjoy being admired by men.

Thank you for the suggestion. We have added women’s enjoyment of their sexualization as a potential avenue for future study in the Individual differences section of the discussion.

---

## [Decision Letter · Decision Letter 1]

31 May 2023

PONE-D-21-22853R1Brains Over Beauty: A Preregistered Test of the Effects of Objectification on Women’s Cognitive PerformancePLOS ONE

Dear Dr. Zola,

thank you for submitting your revised manuscript to PLOS ONE.  Now, as the revised version has been assessed by three experts in the field, we feel that it has merit but does not fully meet PLOS ONE’s publication criteria as it currently stands. Therefore, we invite you to submit a revised version of the manuscript that addresses all the technical and methodological points raised by Reviewer 1.  Please make sure to carefully respond to the Reviewer's comments in  a point-by-point manner.

Please submit your revised manuscript within six months from this date as afterwards, any revision has to be regarded as a new submission. If you will need more time than this to complete your revisions, please reply to this message or contact the journal office at plosone@plos.org. Please include the following items when submitting your revised manuscript:A rebuttal letter that responds to each point raised by the academic editor and reviewer(s). You should upload this letter as a separate file labeled 'Response to Reviewers'.A marked-up copy of your manuscript that highlights changes made to the original version. You should upload this as a separate file labeled 'Revised Manuscript with Track Changes'.An unmarked version of your revised paper without tracked changes. You should upload this as a separate file labeled 'Manuscript'.

Thank you for choosing PLOS ONE for reporting your work.  We look forward to receiving your revised manuscript.

Kind regards,

Sasha

Alexander N. 'Sasha' Sokolov, Ph.D.

Academic Editor

PLOS ONE

Reviewers' comments:

Reviewer's Responses to Questions

**Comments to the Author**

1. If the authors have adequately addressed your comments raised in a previous round of review and you feel that this manuscript is now acceptable for publication, you may indicate that here to bypass the “Comments to the Author” section, enter your conflict of interest statement in the “Confidential to Editor” section, and submit your "Accept" recommendation.

Reviewer #1: (No Response)

Reviewer #2: All comments have been addressed

Reviewer #3: All comments have been addressed

2. Is the manuscript technically sound, and do the data support the conclusions?

Reviewer #1: No

Reviewer #2: Yes

Reviewer #3: Yes

3. Has the statistical analysis been performed appropriately and rigorously? 

Reviewer #1: Yes

Reviewer #2: Yes

Reviewer #3: Yes

4. Have the authors made all data underlying the findings in their manuscript fully available?

Reviewer #1: Yes

Reviewer #2: Yes

Reviewer #3: Yes

5. Is the manuscript presented in an intelligible fashion and written in standard English?

Reviewer #1: Yes

Reviewer #2: Yes

Reviewer #3: Yes

6. Review Comments to the Author

Reviewer #1: I had reviewed this manuscript approx 6 months ago, and while I applauded the attempt to provide better-powered studies to this important question, I pointed to three major flaws. In this new version, none of these have been addressed.

1. In response to the first flaw, the authors write "Although moderators can certainly be important to consider when testing theoretically derived hypotheses, our reading of the published literature on this topic does not lead us to conclude that TSO is an essential moderator when it comes to the effect of objectification on cognitive performance."

Such a reading is inaccurate. The fact that the authors go on to detail the studies that have and have not looked at the moderating effects of TSO does not add anything to what I think is an insufficient answer.

The whole point of the work that is being reviewed here is to provide better empirical tests and to clarify outstanding issues, in the sense of mixed findings. All the literature on objectification points to the importance of predispositions, why not included such measures? Especially since they model their study after the study by Gay and Castano, which finds the interaction effect that is predicted by the objectification theory . I was and remain puzzled by their choice.

2. In response to this point, the authors write "..instead of having a male vs. female experimenter video record participants, we opted to use an anticipated male gaze manipulation as in Calogero et al. (2004), as Calogero et al. found that anticipating a male gaze increased social physique anxiety (a construct that overlaps with body surveillance). Across two studies, Gay and Castano did not include a manipulation check (i.e., a measure of body surveillance/state self-objectification), so it is difficult to evaluate the extent to which their precise manipulation worked – especially since they found no main effects of their male vs. female experimenter manipulation. Gay and Castano did report some statistically significant effects for tests of two-way interactions and a three-way interaction (with TSO and item difficulty), but one interaction suggested that the female experimenter (vs. the male experimenter) had a more negative effect on latency. Given the very low sample sizes in Gay and Castano (25 participants in one study and 50 in another), it is difficult to know what to make of these interactions and the likelihood of false positive findings in these analyses appears high."

Again, the immediate question is WHY? Calogero et al used anticipated male gaze, but they were looking at anxiety, which is by definition a response to something that MAY or WILL happen. The focus of this experiment is not the same. The comments regarding the lack of a manipulation check in the experiments by Gay and Castano is not relevant. The present experiment was set up to IMPROVE upon previous studies, including that by Gay and Castano. This is clear from the beginning, but it should not be used as a justification for using a ì manipulation that is inappropriate given the goal of the experiment. Finally, the comment "especially since they found no main effects of their male vs. female experimenter manipulation. " is bizarre, since it is an interaction of the manipulation with TSO that objectification theory predicts (see #1).

3. I find the response to the issue of item difficulty also unsatisfactory. Easy items are easy items. Given previous findings, one would expect that items with varying difficulty be chosen, so to be able to add difficulty as a predictor. In fact, in the study by Gay and Castano, this was done in both experiments, and in both experiments the pattern of results indicates clearly that difficulty needs to be taken into consideration. In this experiments the authors, who allegedly aim to provide a more sound test than what was done in previous experiments, not only do not vary item difficulty, but chose only simple ones. Puzzling.

In addition, I re-read carefully the procedure, and I find the set up quite strange and extremely artificial. The fact that participants had to stand at a specific place, use an ipad, and give their answer to questions their heard through an audiorecording on a whiteboard. At the very least, this procedure makes the response latency data unusable. Experiments measuring latency do so recording item by item latency, through a software that is purpose-built. And this was done in the experiments on objectification as well, to which the authors refer. Why using such an unreliable procedure that is all but guaranteed to provide inaccurate latencies?

In conclusion, the authors are rightly interested in testing these hypotheses using the new standards that, in the last 10 years or so, researchers have agreed upon to improve the quality of our research and thus confidence in the results that are published. The study presented here, however, suffers from major, fatal flaws. The authors could and should have done better, precisely because there is so much previous research that informs their current choices.

In my view, an article which is predicated upon improving the quality of experimental test of hypotheses derived from objectification theory should careful test these hypotheses, also building on the strengths of and insights gained from, the very previous experiments that they want to improve upon. This experiment does not.

Reviewer #2: Thank you for the opportunity to once again review this work. I think the authors did a wonderful job addressing the concerns outlined by me and the other two reviewers. I look forward to seeing this paper in press!

Reviewer #3: The authors carefully addressed all the issues raised by the reviewers (including myself). In my judgement, the paper is ready to be published and it has the potential to advance current understanding of the effects of self-objectification on women's cognitive performance.

I have only one minor comment: Although they use this term, Quinn et al.'s (2006) study did not implement "a modified version" of the Stroop test. Rather, it used a flawed calculation of the DV (looking at participants' reaction times, instead of interference score, is simply a wrong way to evaluate their' "allocation of attentional resources"). Eronen and Bringmann (2022) argue that one of the barriers to the advancement of our field is that even when findings are demonstrated to be flawed and/or non-replicable, scholars continue to cite them. I therefore recommend mentioning, maybe in a footnote, or in Table 1 (which summarizes what was done in previous research), that the conclusions derived from Quinn et al.'s (2006) study are questionable.

7. PLOS authors have the option to publish the peer review history of their article (what does this mean?). If published, this will include your full peer review and any attached files.

Reviewer #1: No

Reviewer #2: No

Reviewer #3: No

---

## [Author Response · Author response to Decision Letter 1]

21 Aug 2023

Detailed Responses to Reviewer Comments

1. Reviewer 1 insists that trait self-objectification (TSO) is an essential moderator, and that this study is not publishable because we did not assess TSO. Reviewer 1 writes, “Previous research, discussed by these authors, finds that the effects of an objectification manipulation is likely moderated by trait self-objectification. Objectification scholars clearly theorize this.”

We fundamentally disagree with Reviewer 1’s claim that Objectification Theory makes the prediction that the effects of interpersonal objectification rely on an interaction with trait-level objectification. A careful, line-by-line reading of Fredrickson and Roberts’ (1997) Objectification Theory reveals no such claim. Indeed, as we thoroughly review in our manuscript, many authors have found cognitive effects of objectification manipulations without considering an interaction with TSO (pp. 36-38). It is clear that many scientists view these effects as robust enough that they do not require TSO as a covariate in order to be measurable. This issue is far from settled in the scientific literature.

In our manuscript, we carefully outline the literature concerning the role of TSO in objectification’s effect on performance, for example, noting, “Four studies [2,14,19,23] found no evidence that trait self-objectification moderates the effect of objectification on performance. Of these, one reported a significant main effect of objectification on performance using the cut-off p < .05 [23], and three did not [2,14,19]. Kahalon et al. (Study 1, [24]) found trait self-objectification moderated the effect of condition (i.e., receiving an appearance compliment) on math test scores, but did not replicate this finding in the paper’s second study.”(p.37) In summary, some previous studies found that objectification manipulations increase self-objectification and impair cognitive performance regardless of trait self-objectification. Other studies have only been able to show these effects when using trait self-objectification as a moderator. Still others found no evidence for trait self-objectification as a moderator. In other words, based on empirical findings, it is not clear that researchers must control for trait-level self-objectification to see the effects of objectification manipulations. We have updated this section of the paper to include the recent study by Savage and Couture Bue (2023), which also found TSO did not moderate the relationship between state self-objectification and cognitive performance. 

In response to this revision, Reviewer 1 wrote, “Such a reading is inaccurate. The fact that the authors go on to detail the studies that have and have not looked at the moderating effects of TSO does not add anything to what I think is an insufficient answer…Especially since they model their study after the study by Gay and Castano, which finds the interaction effect that is predicted by the objectification theory.”

First, we did not model our study on Gay and Castano. Further, as we noted in our revision letter, the role of TSO as a moderator in Gay and Castano was inconsistent across two studies. In Study 1 (n = 25), the authors did find a TSO by condition interaction on letter-number sequencing (LNS) latency, whereby participants with higher TSO performed worse in the condition with the male experimenter compared to the condition with the female experimenter (p = .02). The same TSO by condition interaction did not reach statistical significance in Study 2 with 50 participants (p = .07).

2. Reviewer 1 remains concerned about the nature of the manipulation in our study. In the first round of revisions, Reviewer 1 wrote, “The authors seem to model their manipulation after the study by Gay and Castano, in which the above-mentioned interaction was found. Yet, instead of having women in the male-gaze condition filmed by a man, they only tell them that the video might be later viewed by a man. Why the authors chose to water-down so much a manipulation that not only had worked in the past, but that it is also a direct operationalization of the theory of objectification and it has excellent ecological validity, is puzzling to me.”

It is true that multiple elements of our methodology differ from those employed by Gay and Castano. Our paper is by no means an attempt at a direct replication of this specific study and at no point have we claimed that it is. To help clarify, we have added substantial detail to our paper explaining how and why we chose the manipulations we did (p.14-15). 

Though Reviewer 1 praises the strength of the manipulation of using a male vs. female experimenter (as in Gay & Castano), we do not find the results of the Gay and Castano paper to be a persuasive demonstration of the superiority of this particular method of manipulating interpersonal objectification. Gay and Castano did not include a manipulation check and in both studies, condition had no significant effects on two of the three DVs (even when TSO and difficulty were taken into account). Without more evidence, our confidence that the male vs. female experimenter manipulation effectively increased women’s state self-objectification is tempered. 

Reviewer 1 suggests that we modeled our manipulation after the one used in Gay and Castano, however, in Gay and Castano participants were video recorded (by either a male or female experimenter) from the neck-down while walking up and down a hallway for 2 minutes. The participants then watched video recording in its entirety before completing a LNS task on a computer and a GMAT math test. In our study, participants were video recorded from the neck-down while simultaneously completing the cognitive task (an attention task modeled after a different section of the WAIS, not the LNS). Participants did not review the video at any point during the study. 

We based the design of our studies on a thorough review of this body of literature, attempting to capitalize on the strengths of different approaches in different papers. We are unclear as to why Reviewer 1 is so highly focused on Gay and Castano’s manipulation, despite the methodologically diverse body of literature on this topic. 

Nonetheless, we took care to note the reviewer’s concerns in our response and in the revised discussion section (p. 31). From our response to the previous round of reviews: “In our discussion section, we acknowledge that despite its efficacy in Calogero et al. (2004), it is possible the anticipated male gaze manipulation was not strong enough and that this could explain why we did not see significantly different levels of body surveillance between our two experimental conditions (and found significant differences only between our experimental conditions and our control). Nonetheless, we observed no effect of condition on performance despite a moderate effect of the control versus the experimental conditions on body surveillance; we focus on results between experimental conditions and control rather than differences between the two experimental conditions.”

3. Reviewer 1 suggests that item difficulty should have been taken into account in analyses. Reviewer 1 writes, “In this [sic] experiments the authors, who allegedly aim to provide a more sound test than what was done in previous experiments, not only do not vary item difficulty, but chose only simple ones. Puzzling.”

We explained this choice of outcome measure in the manuscript (see pages 33-34) and in responses to the first round of requested revisions. In our manuscript we note:

 “The simplicity of the problems was also intended to avoid stereotype threat effects, which are more likely to emerge on difficult tasks (for a review see Pennington et al.[36]). This was especially important as stereotype threat effects have been raised as an alternate explanation for women’s reduced performance on advanced math tests in objectification studies[2,22,24]” (p. 33).“Even simple arithmetic operations require substantial working memory and could be disrupted by other cognitive demands[65], such as body monitoring. Moreover, math problems are frequently employed as distractor tasks in experiments because they are cognitively tasking and load working memory (e.g.,[66])” (p. 34).

“Consistent with this goal, our results showed substantial variability in the time taken to complete the task, even among participants with perfect accuracy” (p. 34).

Reviewer 1 additionally responded in the second round of reviews, “In fact, in the study by Gay and Castano, this was done in both experiments, and in both experiments the pattern of results indicates clearly that difficulty needs to be taken into consideration.” 

Again, we strongly disagree with the characterization that our study uses, or was designed to imitate, the methodology used in Gay and Castano. Nonetheless, we carefully reviewed Gay and Castano’s results in terms of item difficulty. Gay and Castano used a Letter Number Sequencing (LNS) task (accuracy and latency) and an advanced math test as DVs. It is unclear in the paper whether item difficulty was taken into account when analyzing the math test. Regardless, in Study 1, the authors report no statistically significant main effects or interactions in terms of the math test, and drop this DV from Study 2. The LNS task used in the paper is somewhat more similar to the DV employed in our study. Gay and Castano qualified their two-way interaction of TSO (high, low) and condition (male experimenter, female experimenter) with a continuous measure of item difficulty on the LNS task. In Studies 1 (n = 25) and 2 (n = 50), the authors found that for difficult items, participants with high TSO took longer in the male versus female experimenter condition. However, in Study 1 the same three-way interaction revealed that for difficult items, participants with low TSO were slower after being videoed by a female versus male experimenter. It is not clear why participants with low trait self-objectification would perform worse on difficult (versus easy) items after being videoed by a female (the “low” objectification condition) versus male experimenter (the “high” objectification condition). In terms of LNS performance, the authors find a TSO by difficulty interaction in Study 1, but no significant two- or three-way interactions in Study 2. Especially without a control condition, these results are difficult to interpret with confidence. Further, our confidence in the strength of statistically significant results from three-way interactions with very small sample sizes remains limited. Additionally, virtually no published papers on the cognitive effects of interpersonal objectification have examined item difficulty in this way. It is far from the norm with respect to empirical or theoretical approaches in this literature. 

In sum, Reviewer 1 seems to argue that interpersonal objectification will have cognitive effects on women only if a) their TSO is taken into account and b) item difficulty is taken into account. As evident in Table 1 of our paper (which provides a summary of published studies on this topic, p.10-12), this does not appear to be true.

---

## [Editor Report · Decision Letter 2]

7 Sep 2023

Brains Over Beauty: A Preregistered Test of the Effects of Objectification on Women’s Cognitive Performance

PONE-D-21-22853R2

Dear Dr. Zola,

We’re pleased to inform you that your manuscript has been judged suitable for publication and will be formally accepted for publication once it meets all outstanding technical requirements.

Thank you for choosing PLOS ONE for reporting your work.Kind regards,

SashaAlexander N. 'Sasha' Sokolov, Ph.D.

Academic Editor

PLOS ONE
---

## [Editor Report · Acceptance letter]

13 Sep 2023

PONE-D-21-22853R2 

Brains Over Beauty: A Preregistered Test of the Effects of Objectification on Women’s Cognitive Performance 

Dear Dr. Zola:

I'm pleased to inform you that your manuscript has been deemed suitable for publication in PLOS ONE. Congratulations! Your manuscript is now with our production department. 

Kind regards, 

on behalf of

Dr. Alexander N. Sokolov 

Academic Editor

PLOS ONE